# Point3R: Streaming 3D Reconstruction with Explicit Spatial Pointer Memory

Yuqi Wu[1,*]   Wenzhao Zheng[1,*,†]   Jie Zhou[1]   Jiwen Lu[1,2]

[1]Department of Automation, Tsinghua University
[2]Beijing National Research Center for Information Science and Technology
https://ykiwu.github.io/Point3R/

## Abstract

Dense 3D scene reconstruction from an ordered sequence or unordered image collections is a critical step when bringing research in computer vision into practical scenarios. Following the paradigm introduced by DUSt3R, which unifies an image pair densely into a shared coordinate system, subsequent methods maintain an implicit memory to achieve dense 3D reconstruction from more images. However, such implicit memory is limited in capacity and may suffer from information loss of earlier frames. We propose Point3R, an online framework targeting dense streaming 3D reconstruction. To be specific, we maintain an explicit spatial pointer memory directly associated with the 3D structure of the current scene. Each pointer in this memory is assigned a specific 3D position and aggregates scene information nearby in the global coordinate system into a changing spatial feature. Information extracted from the latest frame interacts explicitly with this pointer memory, enabling dense integration of the current observation into the global coordinate system. We design a 3D hierarchical position embedding to promote this interaction and design a simple yet effective fusion mechanism to ensure that our pointer memory is uniform and efficient. Our method achieves competitive or state-of-the-art performance on various tasks with low training costs. Code: https://github.com/YkiWu/Point3R.

## 1  Introduction

Dense 3D reconstruction from image collections has long been a fundamental task in computer vision, with broad applications in fields such as autonomous driving [25, 78], medical modeling, and cultural heritage preservation. Conventional approaches [1, 11, 48, 56, 67, 68] first conduct a sequence of sub-tasks, including feature extraction and matching [33, 41], triangulation, and global alignment to get sparse geometry and camera poses. Multi-view stereo [20, 22, 49] is then used to obtain dense geometry. However, this tightly coupled pipeline is inefficient and tends to be vulnerable to noise. To address these challenges, DUSt3R [64] proposes a data-driven paradigm that directly reconstructs the input image pair as point maps within a unified coordinate system.

Due to the constraint of pair-wise input, DUSt3R requires an additional global alignment step when performing dense reconstruction from multiple images, which is inefficient in multi-view settings. To improve this, subsequent works [59, 60, 63, 71] can be categorized into two main paradigms. One category of these works [60, 71] takes all input images simultaneously and employs global attention to reconstruct them into a unified coordinate system, requiring substantial computational resources and is misaligned with the incremental nature of real-world reconstruction scenarios. The second category [59, 63] introduces an external memory mechanism to retain information from past frames,

---

*Equal contribution. †Corresponding author.

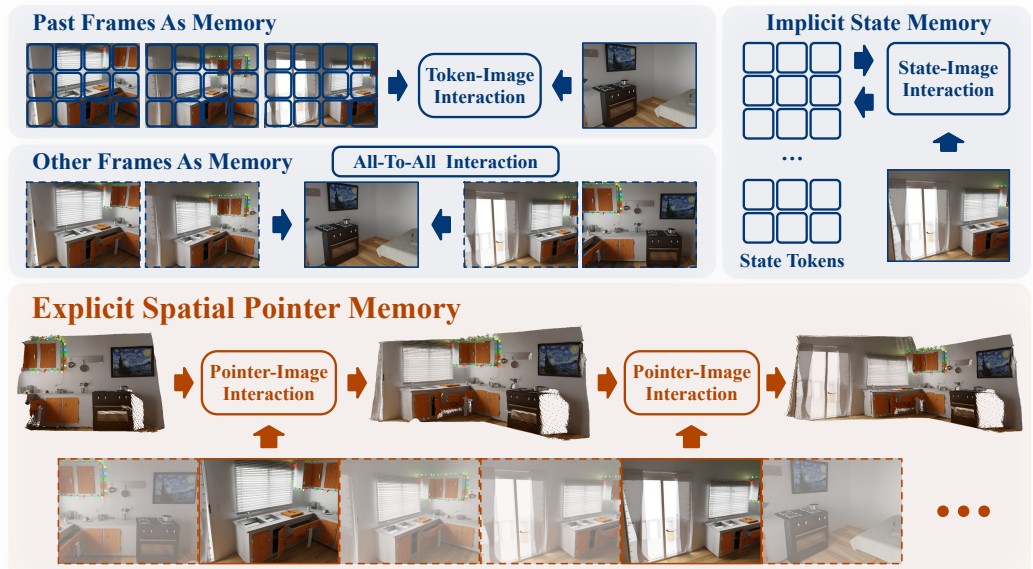

Figure 1: **Comparison between our explicit spatial pointer memory and other paradigms in dense 3D reconstruction.** Methods that conduct all-to-all interaction among all inputs simultaneously [60, 71] can be considered as using **other frames as memory** (for one of the inputs). Methods that cache encoded features of processed frames and conduct token-image interaction [59] can be considered as using **past frames as memory**. Methods maintaining a fixed-length state memory and conducting state-image interaction [63] can be considered as using **implicit state memory**. We propose an **explicit spatial pointer memory** in which each pointer is assigned a 3D position and points to a changing spatial feature. We conduct a pointer-image interaction to integrate new observations into the global coordinate system and update our spatial pointer memory accordingly.

enabling each new input to be directly integrated into the global coordinate system. For instance, Spann3R [59] maintains a memory that essentially caches implicit features of processed frames. However, this implicit memory often contains redundant information, and a discard strategy must be employed once capacity is saturated. CUT3R [63] uses a fixed-length token-based memory mechanism and directly updates it through interactions with image features. Nonetheless, as the number of processed frames increases, this memory inevitably leads to the loss of earlier information.

Inspired by the human memory mechanism, we propose **Point3R**, an online framework equipped with a spatial pointer memory. Human memory of previously encountered environments is inherently related to spatial locations. For example, when we talk about a café or our bedroom, the images we recall are distinct. Similarly, each 3D pointer in our spatial pointer memory is assigned a 3D position in the global coordinate system. Each 3D pointer is directly linked to an explored spatial location and points to a dynamically updated spatial feature. Leveraging this 3D-aware structure, we introduce a 3D hierarchical position embedding, which is integrated into the interaction module between current image features and stored spatial features, enabling more efficient and structured memory querying. Furthermore, since the spatial pointer memory expands as the scene exploration progresses, we design a simple yet effective fusion mechanism to ensure that the memory remains spatially uniform and efficient. Our spatial pointer memory evolves in sync with the current scene, allowing our model to handle both static and dynamic scenes. We use Figure 1 to compare our spatial pointer memory with other paradigms mentioned before. Our method achieves competitive or state-of-the-art performance across various tasks: dense 3D reconstruction, monocular and video depth estimation, and camera pose estimation. It is worth mentioning that although trained on a variety of datasets, the training of our method has a low cost in time and computational resources.

## 2   Related Work

**Conventional 3D Reconstruction.** Classic approaches to 3D reconstruction from image collections are typically optimization-based and tailored to specific scenes. Structure-from-motion

(SfM) [1, 11, 48, 56, 67, 68] follows a pipeline consisting of feature extraction [16, 34, 46], image matching [9, 31, 50, 68], triangulation to 3D, and bundle adjustment [2, 58] to obtain sparse geometry and estimated camera poses. Building upon this, subsequent methods such as Multi-view Stereo (MVS) [18, 19, 49, 65], Neural Radiance Fields (NeRF) [4, 8, 36, 37, 62], and 3D Gaussian Splatting (3DGS) [26] leverage known camera parameters to recover dense geometry or high-fidelity scene representation. These approaches rely on a sequential combination of multiple modules, which not only require considerable optimization time but are also vulnerable to noise. It is worth noting that Simultaneous Localization and Mapping (SLAM) [13, 17, 27, 40] can perform localization and reconstruction in an online manner. However, they often rely on specific camera motion assumptions [13, 40] (sometimes these motion assumptions may be misleading or restrictive) or require additional depth/LiDAR sensors [39] for better performance.

**Learning-Based 3D Reconstruction.** To enhance accuracy and efficiency, the field of 3D reconstruction is gradually shifting toward learning-based and end-to-end paradigms. Some approaches utilize learnable modules to replace hand-crafted components [14, 47] during the traditional reconstruction pipeline. Some others try to optimize the overall pipeline in an end-to-end manner [57, 61, 72]. DUSt3R [64] introduces a pointmap representation and directly learns to integrate an image pair into the same coordinate system, which unifies all sub-tasks we have mentioned above. MonST3R [75] extends this paradigm to dynamic scenes by fine-tuning it on dynamic datasets. However, this pair-wise formulation needs a global alignment to process more views, which is computationally intensive and time-consuming. Subsequent works [59, 60, 63, 71] are exploring how to further replace the global optimization step with an end-to-end learning framework. These efforts can be broadly categorized into two main branches. The former [60, 71] feeds all images simultaneously and leverages global attention to reconstruct the scene within a unified coordinate system; the latter [59, 63] proposes a streaming paradigm, in which a memory module stores information from past frames, thereby enabling online incremental reconstruction from sequential inputs. The streaming paradigm aligns more closely with real-world applications, offering improved scalability without imposing excessive computational demands.

**Streaming Reconstruction and Memory Mechanism.** The streaming reconstruction paradigm and the memory mechanism are inherently aligned with each other. Existing streaming reconstruction methods [10, 17, 59, 63, 76] universally incorporate a certain form of memory to store information from past frames. This memory can take on various forms, such as explicit scene representation (the most direct form of memory), recurrent neural network architectures [10], and encoded or learnable token features [59, 63]. Explicit 3D scene representation is compact and efficient but tailored to a specific scene, limiting its generalizability. Spann3R [59] stores implicit features from previous frames in the memory, which may lead to redundancy. CUT3R [63] employs a fixed-length set of learnable token features as its memory module, which is continuously updated during sequential processing. However, its limited capacity can lead to information loss. In contrast, we propose a spatial pointer memory, in which each pointer is dynamically assigned a 3D position. This ensures that the total amount of stored information scales naturally with the extent of the explored scene. Furthermore, each pointer has a spatial feature that captures aggregated scene information nearby.

## 3   Proposed Approach

### 3.1   Memory-Based Streaming 3D Reconstruction

To densely reconstruct image collections $\mathbf{I} \in \mathbb{R}^{N \times H \times W \times 3}$ into a unified global coordinate system as per-frame pointmaps $\mathbf{X} \in \mathbb{R}^{N \times H \times W \times 3}$, existing methods can be categorized into three paradigms: (1) pair-wise reconstruction [29, 64, 75] with an optimization-based global alignment, (2) one-shot global reconstruction [60, 71] with all inputs, and (3) frame-by-frame input and reconstruction [59, 63] based on a memory mechanism. Pair-wise methods take only one image pair each time as input and reconstruct it into a local coordinate system. A global alignment is then conducted to merge all local outputs into a global coordinate system. This approach suffers from inefficiency due to the need for repeated pair-wise processing and a post-processing stage. The second category feeds all images into the model simultaneously and employs a global attention to reconstruct them directly in a shared coordinate system. However, this approach is computationally intensive and inherently mismatched with the core demands of embodied agents, which typically perform streaming perception of their surroundings and respond correspondingly in practical scenarios.

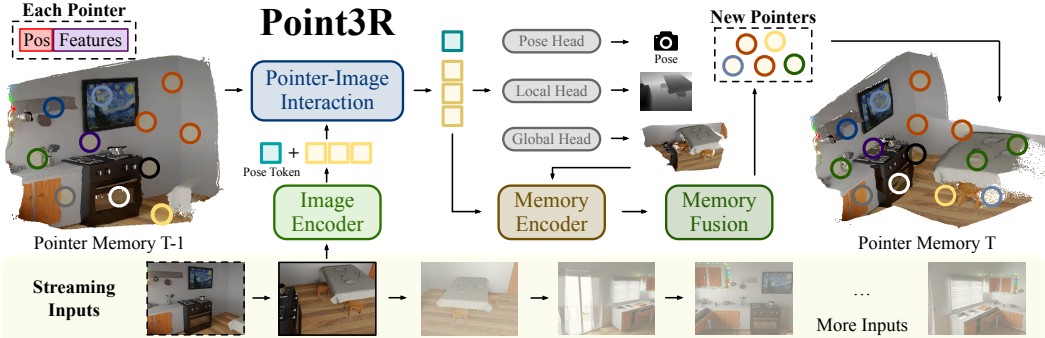

Figure 2: **Overview of Point3R.** Given streaming image inputs, our method maintains an explicit spatial pointer memory to store the observed information of the current scene. We use a ViT [15, 64] encoder to encode the current input into image tokens and use ViT-based decoders to conduct interaction between image tokens and spatial features in the memory. We use two DPT [43] heads to decode local and global pointmaps from the output image tokens. Besides, a learnable pose token is added during this stage so we can directly decode the camera parameters of the current frame. Then we use a simple memory encoder to encode the current input and its integrated output into new pointers, and use a memory fusion mechanism to enrich and update our spatial pointer memory.

In pursuit of a balance between practicality and efficiency, the key idea of memory-based methods is maintaining a memory that stores observed information and interacts with each incoming frame to enable streaming dense reconstruction. We formulate the memory-based pipeline as follows:

$$X_t = \mathcal{F}(\mathcal{M}_{t-1}, I_t),\tag{1}$$

where $I_t \in \mathbb{R}^{H \times W \times 3}$ and $X_t \in \mathbb{R}^{H \times W \times 3}$ are the current image input and pointmap output, $\mathcal{F}$ is a certain approach and $\mathcal{M}_{t-1}$ represents a memory with integrated information from the past frames.

To elaborate, in Spann3R [59], $\mathcal{M}$ is a growing set of key-value-pair features, where the features are encoded from the output of each previous frame. In CUT3R [63], $\mathcal{M}$ takes the form of a fixed-length feature sequence that is iteratively updated as new frame comes. In this work, we propose an explicit spatial pointer memory $\mathcal{M}$ that stores a set of 3D pointers corresponding to the explored regions of the current scene, along with their associated spatial features. We argue that this explicit memory enables compact and structured integration of information from past observations. Each pointer is indexed by a 3D position in the global coordinate system, rather than implicit features, making the interaction between the memory and the current frame more direct and efficient.

### 3.2 Pointer-Image Interaction

The core of our method is making the interaction between the current input and our spatial pointer memory more effective and efficient. We will elaborate on the overall model architecture, which is composed of an image encoder, interaction decoders, a memory encoder, and a memory fusion mechanism. We also incorporate a 3D hierarchical position embedding into the interaction decoders to promote our pointer-image interaction. Figure 2 shows the overview of our method.

**Image Encoder.** For each frame, we use a ViT [15] to encode the current input $I_t$ into image tokens $F_t$:

$$F_t = \text{Encoder}(I_t).\tag{2}$$

**Interaction Decoders.** Our spatial pointer memory $\mathcal{M}$ consists of a set of 3D pointers (3D positions $P$ and spatial features $M$). Before processing the first frame, the memory has not yet stored any global spatial features of the current scene. Therefore, we use a simple layer to embed the image tokens $F_0$ of the first frame, and use the output $M_0$ to initialize the features of our memory. It is worth noting that, since the precise spatial positions represented by these features are not yet known at this time, each feature has not been assigned a specific 3D position. Then we use two intertwined decoders [64, 66] to enable interaction between the current image tokens and the memory:

$$F_0', z_0' = \text{Decoders}((F_0, z_0), M_0),\tag{3}$$

$$F_t', z_t' = \text{Decoders}((F_t, z_t), M_{t-1}),\tag{4}$$

where we use a learnable pose token $z$ as a bridge between the current frame and the global coordinate system. After this interaction, we get the updated image tokens $F'_t$ and pose token $z'_t$. Then we use $F'_t$ and $z'_t$ to predict two pointmaps ($\hat{X}_t^{self}$ in the local coordinate system of the current input and $\hat{X}_t^{global}$ in the global coordinate system) with their own confidence maps ($C_t^{self}$ and $C_t^{global}$), and a camera pose $\hat{T}_t$ representing the rigid transformation between this two coordinate system:

$$\hat{T}_t = \text{Head}_{pose}(z'_t), \tag{5}$$

$$\hat{X}_t^{self}, C_t^{self} = \text{Head}_{self}(F'_t), \tag{6}$$

$$\hat{X}_t^{global}, C_t^{global} = \text{Head}_{global}(F'_t, z'_t), \tag{7}$$

where $\text{Head}_{pose}$ is an MLP network, $\text{Head}_{self}$ and $\text{Head}_{global}$ are DPT [43] heads. The global coordinate system is actually the first input's own coordinate system.

**Memory Encoder.** After processing $I_t$, we use the current features $F_t, F'_t$ and the predicted pointmap $\hat{X}_t^{global}$ to obtain the new pointers:

$$P_{new}(u, v) = \frac{1}{|R_{u,v}|} \sum_{(i,j) \in R_{u,v}} \hat{X}_t^{global}(i, j), \tag{8}$$

$$M_{new} = \text{Encoder}_f(F_t, F'_t) + \text{Encoder}_{geo}(\hat{X}_t^{global}), \tag{9}$$

where $M_{new}$ is the set of new spatial features and we compute the 3D location $P_{new}(u, v)$ for each feature in the global coordinate system as its 3D position by averaging all 3D coordinates within the corresponding patch $R_{u,v}$. Besides, $\text{Encoder}_f$ is a MLP and $\text{Encoder}_{geo}$ is a lightweight ViT.

**Memory Fusion Mechanism.** Apart from the first frame when we simply add all the obtained pointers into the memory, new pointers extracted from each subsequent frame $I_t$ are integrated into the existing memory $\mathcal{M}_{t-1}$ through a fusion mechanism. To elaborate, we compute the Euclidean distance between each new pointer and all existing pointers from the memory to identify its nearest neighbor. If the distance to its nearest neighbor in the memory is below a changing threshold $\delta$ (we change $\delta$ accordingly to make the distribution of memory units more uniform), we treat the neighbors as corresponding and perform the fusion. Otherwise, the new pointer is directly added to the memory. Notably, if a memory pointer is identified as the nearest neighbor by one or a few new pointers, we update the position $p$ and spatial feature $m$ of this pointer as follows:

$$p' = \frac{1}{K} \sum_{i=1}^{K} p_i^{new}, m' = \frac{1}{K} \sum_{i=1}^{K} m_i^{new}, \tag{10}$$

where $K$ is the total number of new neighbors of the target pointer. This fusion mechanism ensures that each pointer always stores the current spatial information at its location, thereby enabling the memory to deal with dynamic scenes. In this way, we obtain an enriched and updated memory $\mathcal{M}_t$.

**3D Hierarchical Position Embedding.** We expand the rotary position embedding (RoPE [23, 54], a method of relative position embedding usually used in transformers) to a 3D hierarchical position embedding and use this to conduct position embedding in continuous 3D space. In practical implementation, RoPE utilizes multiple frequencies $\theta_t$ using channel dimensions $d_{head}$ of key and query as

$$\theta_t = 10000^{-t/(d_{head}/2)}, \text{where } t \in \{0, 1, ..., d_{head}/2\}. \tag{11}$$

Then a rotation matrix $\mathbf{R} \in \mathbb{C}^{N \times (d_{head}/2)}$ is defined as

$$\mathbf{R}(n, t) = e^{i\theta_t n}, \tag{12}$$

and applied to query and key with the Hadamard product $\circ$ as

$$\bar{\mathbf{q}}' = \bar{\mathbf{q}} \circ \mathbf{R}, \ \bar{\mathbf{k}}' = \bar{\mathbf{k}} \circ \mathbf{R}, \ \mathbf{A}' = \text{Re}[\bar{\mathbf{q}}'\bar{\mathbf{k}}'^*]. \tag{13}$$

Here, the attention matrix with RoPE $\mathbf{A}'$ implies relative position in rotation form $e^{i(n-m)\theta_t}$. Inspired by this, we formulate a 3D hierarchical position embedding. We need to change the 1D token index $n$ in RoPE to a 3D token position $p_n = (p_n^x, p_n^y, p_n^z)$ where $p_n^x$, $p_n^y$ and $p_n^z$ correspond to the coordinates on three axes in a continuous 3D space. Thus, the rotation matrix $\mathbf{R} \in \mathbb{C}^{N \times (d_{head}/2)}$ in Eq. 12 is changed accordingly as

$$\mathbf{R}(n, 3t) = e^{i\theta_t p_n^x}, \ \mathbf{R}(n, 3t+1) = e^{i\theta_t p_n^y}, \ \mathbf{R}(n, 3t+2) = e^{i\theta_t p_n^z}. \tag{14}$$

To accommodate spatial position inputs of varying scales, we use $h$ different frequency bases (10000 in Eq. 11) to derive hierarchical rotation matrices and apply them to query and key as

$$\bar{\mathbf{q}}' = \frac{1}{h}\sum_{i=1}^{h}(\bar{\mathbf{q}} \circ \mathbf{R}_i), \ \bar{\mathbf{k}}' = \frac{1}{h}\sum_{i=1}^{h}(\bar{\mathbf{k}} \circ \mathbf{R}_i), \ \mathbf{A}' = \text{Re}[\bar{\mathbf{q}}'\bar{\mathbf{k}}'^*]. \tag{15}$$

We use this hierarchical position embedding in our interaction decoders to inject relative position information into image tokens and memory features. When applying the rotation matrices in Eq. 15, the 3D token position of each memory feature is its 3D position. For image tokens from $I_t$, the 3D token position assigned to each of them is as follows:

$$p(u,v) = \frac{1}{|R_{u,v}|}\sum_{(i,j)\in R_{u,v}} \hat{X}_{t-1}^{global}(i,j), \tag{16}$$

where $R_{u,v}$ is the corresponding image patch and we use $\hat{X}_{t-1}^{global}$ from $t-1$ because we assume that the image tokens of the current frame are more likely to be spatially close to those of the previous frame. Of course, even in cases of significant pose shifts or unordered inputs, this assumption will not introduce any adverse effects. This is because the image tokens of the current frame interact with all memory features in our interaction decoders, and our hierarchical position embedding merely provides a potential prior. Due to the space limit, we will add more details about the design and implementation of our 3D hierarchical position embedding in the supplementary material.

## 3.3 Training Strategy

**Training Objective.** Following MASt3R [29] and CUT3R [63], we use the L2 norm loss for the poses and a confidence-aware loss for the pointmaps. In practical implementation, we parameterize the predicted pose $\hat{T}_t$ as quaternion $\hat{q}_t$ and translation $\hat{\tau}_t$. We use $\hat{\mathcal{X}} = \{\hat{\mathcal{X}}^{self}, \hat{\mathcal{X}}^{global}\}$ to denote the predicted pointmaps, where $\hat{\mathcal{X}}^{self} = \{\hat{X}_t^{self}\}_{t=1}^N$, $\hat{\mathcal{X}}^{global} = \{\hat{X}_t^{global}\}_{t=1}^N$ and $N$ is the number of images per sequence. Besides, $\mathcal{C}$ is used to denote the set of confidence scores correspondingly. So the final expression of the loss we used is:

$$\mathcal{L}_{pose} = \sum_{t=1}^{N}\left(\|\hat{q}_t - q_t\|_2 + \left\|\frac{\hat{\tau}_t}{\hat{s}} - \frac{\tau_t}{s}\right\|_2\right), \tag{17}$$

$$\mathcal{L}_{conf} = \sum_{(\hat{x},c)\in(\hat{\mathcal{X}},\mathcal{C})}\left(c \cdot \left\|\frac{\hat{x}}{\hat{s}} - \frac{x}{s}\right\|_2 - \alpha \log c\right), \tag{18}$$

where $\hat{s}$ and $s$ are scale normalization factors for $\hat{\mathcal{X}}$ and $\mathcal{X}$. When the groundtruth pointmaps are metric, we set $\hat{s} := s$ to force the model to learn metric-scale results.

**Training Datasets.** During training, we use a combination of 14 datasets, including ARKitScenes [5], ScanNet [12], ScanNet++ [74], CO3Dv2 [44], WildRGBD [70], OmniObject3D [69], HyperSim (a subset of it) [45], BlendedMVS [73], MegaDepth [30], Waymo [55], VirtualKITTI2 [7], PointOdyssey [79], Spring [35], and MVS-Synth [24]. These datasets exhibit highly diverse characteristics, encompassing both indoor and outdoor, static and dynamic, as well as real-world and synthetic scenes. See the supplementary material for more details.

**Training Stages.** Our model is trained in three stages. We train the model by sampling 5 frames per sequence in the first stage. The input here is $224\times224$ resolution. Then we use input with different aspect ratios (set the maximum side to 512) in the second stage, following CUT3R [63]. And finally, we freeze the encoder and fine-tune other parts on 8-frame sequences.

**Implementation Details.** We initialize our ViT-Large [15, 64] image encoder, ViT-Base interaction decoders [64, 66], and DPT [43] heads with pre-trained weights from DUSt3R [64]. Our memory encoder is composed of a light-weight ViT (6 blocks) and a 2-layer MLP. Each memory feature has a dimensionality of 768. We use the AdamW optimizer [32] and the learning rate warms up to a maximum value of 5e-5 and decreases according to a cosine schedule. We train our model on 8 A800 NVIDIA GPUs for 15 days, which is a relatively low cost.

Table 1: **Quantitative 3D reconstruction results on 7-scenes and NRGBD datasets.** We use "GA" to mark methods with global alignment, and use "Optim." and "Onl." to distinguish between optimization-based and online methods [63]. Our method achieves competitive or better performance than those optimization-based methods and current online methods.

| Method | Optim. | Onl. | 7-scenes Acc↓ Mean | Med | Comp↓ Mean | Med | NC↑ Mean | Med | NRGBD Acc↓ Mean | Med | Comp↓ Mean | Med | NC↑ Mean | Med |
|---|---|---|---|---|---|---|---|---|---|---|---|---|---|---|
| DUSt3R-GA [64] | ✓ | | 0.146 | 0.077 | 0.181 | 0.067 | 0.736 | 0.839 | 0.144 | **0.019** | 0.154 | **0.018** | **0.870** | **0.982** |
| MASt3R-GA [29] | ✓ | | 0.185 | 0.081 | 0.180 | 0.069 | 0.701 | 0.792 | 0.085 | 0.033 | **0.063** | 0.028 | 0.794 | 0.928 |
| MonST3R-GA [75] | ✓ | | 0.248 | 0.185 | 0.266 | 0.167 | 0.672 | 0.759 | 0.272 | 0.114 | 0.287 | 0.110 | 0.758 | 0.843 |
| Spann3R [59] | | ✓ | 0.298 | 0.226 | 0.205 | 0.112 | 0.650 | 0.730 | 0.416 | 0.323 | 0.417 | 0.285 | 0.684 | 0.789 |
| CUT3R [63] | | ✓ | 0.126 | 0.047 | 0.154 | 0.031 | 0.727 | 0.834 | 0.099 | 0.031 | 0.076 | 0.026 | 0.837 | 0.971 |
| **Ours** | | ✓ | **0.085** | **0.046** | **0.087** | **0.030** | **0.739** | **0.854** | **0.077** | 0.030 | 0.069 | 0.027 | 0.835 | 0.971 |

Table 2: **Monocular Depth Evaluation** on NYU-v2 (static), Sintel, Bonn, and KITTI datasets.

| Method | NYU-v2 (Static) Abs Rel ↓ | $\delta<1.25$ ↑ | Sintel Abs Rel ↓ | $\delta<1.25$ ↑ | Bonn Abs Rel ↓ | $\delta<1.25$ ↑ | KITTI Abs Rel ↓ | $\delta<1.25$ ↑ |
|---|---|---|---|---|---|---|---|---|
| DUSt3R [64] | 0.080 | 90.7 | 0.424 | 58.7 | 0.141 | 82.5 | 0.112 | 86.3 |
| MASt3R [29] | 0.129 | 84.9 | **0.340** | **60.4** | 0.142 | 82.0 | **0.079** | **94.7** |
| MonST3R [75] | 0.102 | 88.0 | 0.358 | 54.8 | 0.076 | 93.9 | 0.100 | 89.3 |
| Spann3R [59] | 0.122 | 84.9 | 0.470 | 53.9 | 0.118 | 85.9 | 0.128 | 84.6 |
| CUT3R [63] | 0.086 | 90.9 | 0.428 | 55.4 | 0.063 | **96.2** | 0.092 | 91.3 |
| **Ours** | **0.078** | **92.3** | 0.395 | 55.0 | **0.061** | 94.5 | 0.083 | 94.6 |

## 4 Experiments

**Tasks and Baselines.** We use various 3D/4D tasks (dense 3D reconstruction, monocular depth estimation, video depth estimation, and camera pose estimation) to evaluate our method. We choose DUSt3R [64], MASt3R [29], MonST3R [75], Spann3R [59], and CUT3R [63] as our primary baselines. Among these methods, DUSt3R, MASt3R, and MonST3R take an image pair as input, so an optimization-based global alignment (GA) stage is conducted when dealing with streaming inputs. Both Spann3R and CUT3R have a memory module so they can process an image sequence in an online manner, similar to our method.

### 4.1 3D Reconstruction

We evaluate the 3D reconstruction performance on the 7-scenes [51] and NRGBD [3] datasets in Table 1, and the metrics we used include accuracy (Acc), completion (Comp), and normal consistency (NC), following previous works [3, 59, 64]. We use inputs with minimal overlap [63]: 3 to 5 frames per scene for the 7-scenes datasets and 2 to 4 frames per scene for the NRGBD dataset. Such sparsely sampled inputs can directly demonstrate the effectiveness of our proposed pointer memory, which is explicitly associated with 3D spatial locations and does not rely on similarity or continuity between input frames. As shown in Table 1, our method achieves comparable or better results than other memory-based online approaches or even DUSt3R-GA [64]. We compare the reconstruction quality of our method with other memory-based online approaches, Spann3R [59] and CUT3R [63] in Figure 3, and our method achieves state-of-the-art reconstruction performance with sparse inputs from the 7-scenes and NRGBD datasets.

### 4.2 Monocular and Video Depth Estimation

**Monocular Depth Estimation.** We evaluate zero-shot monocular depth estimation [63, 75] performance on NYU-v2 [52] (static), Sintel [6], Bonn [42], and KITTI [21] datasets. We adopt per-frame median scaling following DUSt3R [64], and the evaluation metrics we used include absolute relative error (Abs Rel) and percentage of inlier points $\delta < 1.25$. As shown in Table 2, our method achieves state-of-the-art or competitive performance in both static and dynamic, indoor and outdoor scenes.

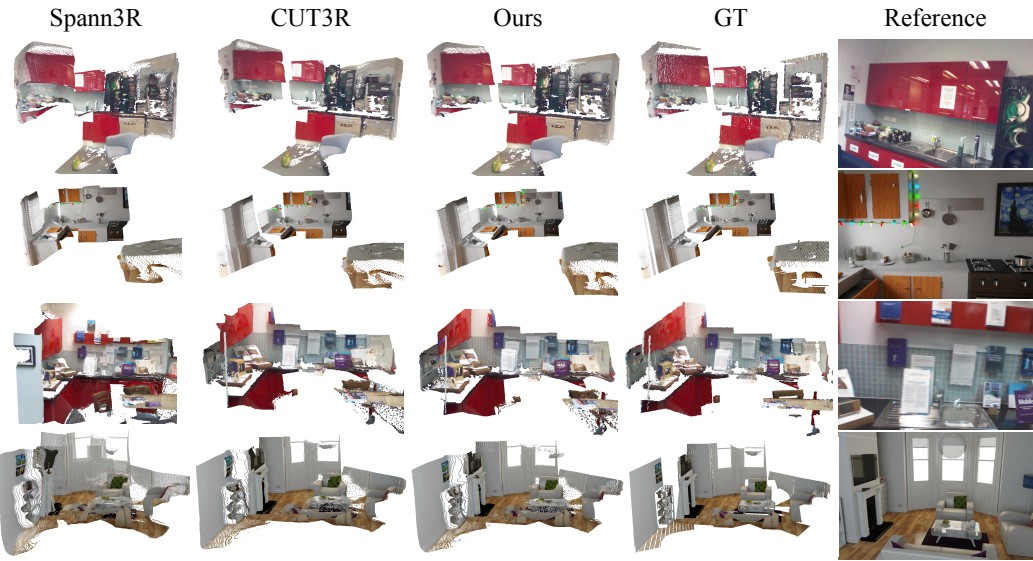

| Spann3R | CUT3R | Ours | GT | Reference |

Figure 3: **Qualitative results on sparse inputs from the 7-scenes and NRGBD datasets.** Our method achieves the best qualitative results among memory-based methods.

Table 3: **Video Depth Evaluation.** We compare scale-invariant depth (per-sequence alignment) and metric depth (no alignment) results on Sintel, Bonn, and KITTI datasets.

| Alignment | Method | Optim. | Onl. | Sintel | | BONN | | KITTI | |
| | | | | Abs Rel ↓ | $\delta < 1.25$ ↑ | Abs Rel ↓ | $\delta < 1.25$ ↑ | Abs Rel ↓ | $\delta < 1.25$ ↑ |
|---|---|---|---|---|---|---|---|---|---|
| Per-sequence | DUSt3R-GA [64] | ✓ | | 0.656 | 45.2 | 0.155 | 83.3 | 0.144 | 81.3 |
| | MASt3R-GA [29] | ✓ | | 0.641 | 43.9 | 0.252 | 70.1 | 0.183 | 74.5 |
| | MonST3R-GA [75] | ✓ | | **0.378** | **55.8** | 0.067 | **96.3** | 0.168 | 74.4 |
| | Spann3R [59] | | ✓ | 0.622 | 42.6 | 0.144 | 81.3 | 0.198 | 73.7 |
| | CUT3R [63] | | ✓ | 0.421 | 47.9 | 0.078 | 93.7 | 0.118 | 88.1 |
| | **Ours** | | ✓ | 0.481 | 44.8 | **0.066** | 95.8 | **0.093** | **93.5** |
| Metric-scale | MASt3R-GA [29] | ✓ | | **1.022** | 14.3 | 0.272 | 70.6 | 0.467 | 15.2 |
| | CUT3R [63] | | ✓ | 1.029 | **23.8** | 0.103 | 88.5 | **0.122** | **85.5** |
| | **Ours** | | ✓ | 1.208 | 13.8 | **0.081** | **95.8** | 0.169 | 80.5 |

**Video Depth Estimation.** We align predicted depth maps to ground truth using a per-sequence scale (Per-sequence alignment) to evaluate per-frame quality and inter-frame consistency. We also compare results without alignment with other metric pointmap methods like MASt3R [29] and CUT3R [63] (Metric-scale alignment). As shown in Table 3, with the per-sequence scale alignment, our method outperforms DUSt3R [64], MASt3R [29], and Spann3R [59] by a large margin. These methods have a static scene assumption and are trained only on static datasets. Our spatial pointer memory directly associates spatial features with their real-world positions and imposes no assumptions or dependencies on whether the scene is static or dynamic. Our method performs comparably, or even better than MonST3R-GA [75] and CUT3R [63] (methods trained on dynamic datasets). Besides, in the metric-scale setting, our method outperforms MASt3R-GA [29] and performs comparably with CUT3R [63], leading on Bonn.

## 4.3 Camera Pose Estimation

We evaluate the camera pose estimation performance on ScanNet [12] (static), Sintel [6], and TUM-dynamics [53] datasets following MonST3R [75] and CUT3R [63]. We report Absolute Translation Error (ATE), Relative Translation Error (RPE trans), and Relative Rotation Error (RPE rot) after applying a Sim(3) Umeyama alignment with the ground truth [75]. It is worth noting that prior approaches conduct additional optimization while our method does not require any post-processing. Results in Table 4 show that our method performs comparably with other online methods, but there persists a performance gap between those optimization-based baselines and our method.

Table 4: **Camera Pose Estimation Evaluation** on ScanNet, Sintel, and TUM-dynamics datasets.

| Method | Optim. | Onl. | ScanNet (Static) | | | Sintel | | | TUM-dynamics | | |
|---|---|---|---|---|---|---|---|---|---|---|---|
| | | | ATE ↓ | RPE trans ↓ | RPE rot ↓ | ATE ↓ | RPE trans ↓ | RPE rot ↓ | ATE ↓ | RPE trans ↓ | RPE rot ↓ |
| Robust-CVD [28] | ✓ | | 0.227 | 0.064 | 7.374 | 0.360 | 0.154 | 3.443 | 0.153 | 0.026 | 3.528 |
| CasualSAM [77] | ✓ | | 0.158 | 0.034 | 1.618 | 0.141 | **0.035** | **0.615** | 0.071 | **0.010** | 1.712 |
| DUSt3R-GA [64] | ✓ | | 0.081 | 0.028 | 0.784 | 0.417 | 0.250 | 5.796 | 0.083 | 0.017 | 3.567 |
| MASt3R-GA [29] | ✓ | | 0.078 | 0.020 | **0.475** | 0.185 | 0.060 | 1.496 | **0.038** | 0.012 | **0.448** |
| MonST3R-GA [75] | ✓ | | **0.077** | **0.018** | 0.529 | **0.111** | 0.044 | 0.869 | 0.098 | 0.019 | 0.935 |
| Spann3R [59] | | ✓ | 0.096 | 0.023 | 0.661 | 0.329 | 0.110 | 4.471 | 0.056 | 0.021 | 0.591 |
| CUT3R [63] | | ✓ | 0.099 | 0.022 | 0.600 | 0.213 | 0.066 | 0.621 | 0.046 | 0.015 | 0.473 |
| **Ours** | | ✓ | 0.097 | 0.035 | 2.791 | 0.442 | 0.154 | 1.897 | 0.058 | 0.031 | 0.758 |

Table 5: **Quantitative 3D reconstruction results on long sequences.**

| Method | 7-scenes | | | | | | NRGBD | | | | | |
|---|---|---|---|---|---|---|---|---|---|---|---|---|
| | Acc↓ | | Comp↓ | | NC↑ | | Acc↓ | | Comp↓ | | NC↑ | |
| | Mean | Med. | Mean | Med. | Mean | Med. | Mean | Med. | Mean | Med. | Mean | Med. |
| CUT3R [63] | 0.238 | 0.172 | 0.105 | 0.025 | 0.527 | 0.537 | 0.372 | 0.270 | 0.211 | 0.090 | 0.556 | 0.582 |
| **Ours** | **0.071** | **0.033** | **0.031** | **0.015** | **0.558** | **0.587** | **0.110** | **0.050** | **0.025** | **0.009** | **0.641** | **0.729** |

## 4.4 Analysis and Discussion

**Comparison on Long Sequences.** We compare our model with CUT3R [63] on long sequences to show the advantage of our explicit spatial pointer memory to effectively store more information from past frames. We resampled the testing sequences from 7-scenes at the interval of 1 (each contains 500-1000 frames) and NRGBD at the interval of 2 (400-900 frames). As shown in the Table 5, our method outperforms CUT3R by a large margin, showing the advantage of our model when handling long sequences in practical applications.

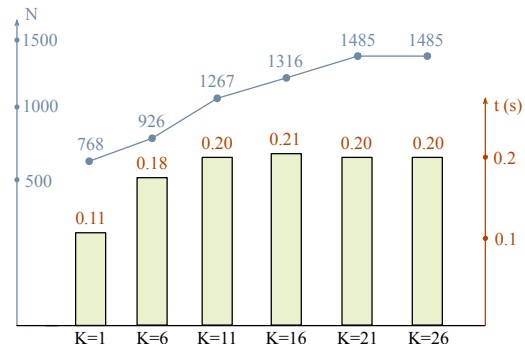

Figure 4: Changes on **the total number of pointers** and **per-frame runtime** with memory fusion.

**Robustness to the Input Orders.** We evaluate the robustness of our model to the order of camera registration. To be specific, for each scene in 7-scenes, we sample a sequence with a frame interval of 20 (each sequence contains 25-50 frames). For each scene in NRGBD, we sample a sequence with a frame interval of 40 (each sequence contains 20-40 frames). Then we disrupt the input sequence after the sampling to evaluate the reconstruction results. From Table 6, we can observe that our method still achieves good reconstruction performance after the input sequence is shuffled. This verifies the robustness of our explicit pointer memory to handle the discontinuity of the inputs.

**Effect of the Memory Fusion Mechanism and Runtime Analysis.** We design the memory fusion mechanism to get a balance between efficiency and performance. Figure 4 shows the changes in the number of pointers N in the memory (line graph) and per-frame runtime t (histogram) with the increasing number of frames K when processing Scene *WhiteRoom* (from NRGBD dataset). We can see that this memory fusion mechanism can control the total number of pointers and per-frame runtime within a reasonable range. We also report the 3D reconstruction results without the memory fusion mechanism on the 7-scenes and NRGBD datasets in Table 7. Although this fusion mechanism may result in a slight decrease in some metrics, we believe that the sacrifice made for efficiency improvement is worthwhile.

**Effect of the 3D Hierarchical Position Embedding.** To demonstrate the effectiveness of our 3D hierarchical position embedding, we conduct ablation studies for this module in Table 7. We report the 3D reconstruction results without the 3D hierarchical position embedding ("w/o 3DHPE"), which shows the effectiveness of the elaborate position embedding we proposed.

**Comparison with SLAM Systems and More Discussions.** Traditional SLAM systems and the recent feed-forward 3D reconstruction methods (e.g., DUSt3R [64], CUT3R [63], and ours) are similar

Table 6: **Robustness to the input orders.**

| Sample | 7-scenes | | | | | | NRGBD | | | | | |
|---|---|---|---|---|---|---|---|---|---|---|---|---|
| | Acc↓ | | Comp↓ | | NC↑ | | Acc↓ | | Comp↓ | | NC↑ | |
| | Mean | Med. | Mean | Med. | Mean | Med. | Mean | Med. | Mean | Med. | Mean | Med. |
| Ordered | 0.032 | 0.016 | 0.024 | 0.008 | 0.665 | 0.758 | 0.064 | 0.031 | 0.029 | 0.011 | 0.801 | 0.949 |
| Shuffled | 0.033 | 0.014 | 0.019 | 0.010 | 0.669 | 0.764 | 0.063 | 0.027 | 0.029 | 0.009 | 0.800 | 0.953 |

Table 7: **Effects of the 3D hierarchical position embedding and the memory fusion mechanism.**

| | 7-scenes | | | NRGBD | | |
|---|---|---|---|---|---|---|
| | Acc↓ | Comp↓ | NC↑ | Acc↓ | Comp↓ | NC↑ |
| Ours (w/o 3DHPE) | 0.142 | **0.132** | 0.698 | 0.083 | 0.079 | 0.808 |
| Ours (w/o Mem-Fusion) | **0.118** | 0.148 | 0.721 | **0.079** | 0.074 | **0.824** |
| Ours | 0.124 | 0.139 | **0.725** | **0.079** | **0.073** | **0.824** |

in terms of output format (reconstruction results and predicted poses), but their core objectives are different. The main objective of SLAM lies in accurately registering the camera poses from RGB-D inputs (where the depth could be obtained from prediction models). Differently, feed-forward 3D reconstruction methods aim to develop a unified model to predict 3D reconstructed points for each frame in a shared coordinate system (and some recent methods simultaneously predict the camera pose with an additional head). To better evaluate these two paradigms, we compare our method with MASt3R-SLAM [38] on its own established benchmarks.

We conduct the dense geometry evaluation on 7-Scenes seq-01 and also report the RMSE of the absolute trajectory error (ATE) in meters. We also compare these two methods in terms of peak memory usage and per-frame runtime. As shown in Table 8, our method achieves better dense geometry reconstruction results with lower memory consumption. However, there is still room for further improvement in tracking accuracy (as discussed in Limitations) and runtime efficiency. We hold the belief that these two paradigms are compatible and essentially beneficial to each other. Visual SLAM systems could use a feed-forward 3D reconstruction model to produce reconstructed points (just like MASt3R-SLAM [38]). Feed-forward 3D reconstruction methods can further employ SLAM techniques such as bundle adjustment to obtain more accurate poses.

Table 8: **Comparison with SLAM in mapping quality, tracking accuracy and efficiency.**

| | Acc↓ | Comp↓ | ATE↓ | Peak Memory Usage↓ | Per-frame Runtime↓ |
|---|---|---|---|---|---|
| MASt3R-SLAM [38] | 0.068 | 0.045 | **0.066** | 7.18 GB | 0.11 s |
| **Ours** | **0.061** | **0.022** | 0.084 | 5.46 GB | 0.20 s |

## 5   Conclusion and Discussions

In this paper, we have presented an online streaming 3D reconstruction framework, Point3R, with a spatial pointer memory. When processing streaming inputs, our method maintains a growing spatial pointer memory in which each pointer is assigned a specific 3D position and aggregates scene information nearby with a changing spatial feature. Equipped with a 3D hierarchical position embedding and a simple yet effective memory fusion mechanism, our method imposes minimal constraints on the input, handling both static and dynamic scenes as well as ordered or unordered image collections. With a low training cost, our method achieves competitive or state-of-the-art performance on various 3D/4D tasks, which verifies the effectiveness of our method.

**Limitations.** As the explored area expands, the positions where pointers are stored also grow progressively, which may introduce additional interference to camera pose estimation in subsequent frames. One of our future works is improving the pointer-image interaction to mitigate this issue.

**Broader Impacts.** Our method facilitates scalable and efficient dense streaming 3D scene reconstruction, benefiting a wide range of applications. The explicit and interpretable design of our pointer memory makes our method more transparent and adaptable to different real-world scenarios.

## Acknowledgements

This work was supported in part by the National Key Research and Development Program of China under Grant 2023YFB280693, in part by the National Natural Science Foundation of China under Grant 62125603, Grant 62336004, Grant 62576188, and Grant 62321005, in part by the Beijing Natural Science Foundation under Grant L247009, and in part by the Beijing National Research Center for Information Science and Technology.

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

# A  More Method Details

## A.1  Memory Fusion Mechanism

We use a changing threshold $\delta$ to determine whether a new pointer and its nearest neighbor are sufficiently close. At time $t$, we have:

$$\delta = \sqrt{(\frac{\max P_{t-1}^x - \min P_{t-1}^x}{l_x})^2 + (\frac{\max P_{t-1}^y - \min P_{t-1}^y}{l_y})^2 + (\frac{\max P_{t-1}^z - \min P_{t-1}^z}{l_z})^2},$$

(19)

where $P_{t-1}^x, P_{t-1}^y, P_{t-1}^z$ are the set of X, Y, Z components of the coordinates of all memory pointers from $\mathcal{M}_{t-1}$, $l_x, l_y, l_z$ are constants we use to control the distribution of memory pointers. In practical implementation, we set $l_x, l_y, l_z$ to 20.

## A.2  3D Hierarchical Position Embedding

For $n$, $m$-th query and key $\mathbf{q}_n, \mathbf{k}_m \in \mathbb{R}^{1 \times d_{head}}$, RoPE converts them to complex vector $\bar{\mathbf{q}}_n, \bar{\mathbf{k}}_m \in \mathbb{R}^{1 \times (d_{head}/2)}$ by considering $(2t)$-th dim as real part and $(2t+1)$-th dim as imaginary part. We follow this so $\theta_t$ in Eq. 14 in the main text is

$$\theta_t = b^{-t/(d_{head}/6)}, \text{where } t \in \{0, 1, ..., d_{head}/6\}, b \in \{10, 100, 1000, 10000\}.$$

(20)

We use different frequency bases $b$ to accommodate spatial inputs of varying scales, and thus derive four ($h$ in Eq. 15 in the main text) different rotation matrices $\mathbf{R}_i$ ($i = 0, 1, 2, 3$). Then we can obtain the embedded query and key, and the corresponding attention matrix as follows:

$$\bar{\mathbf{q}}' = \frac{1}{4}\sum_{i=1}^{4}(\bar{\mathbf{q}} \circ \mathbf{R}_i), \ \bar{\mathbf{k}}' = \frac{1}{4}\sum_{i=1}^{4}(\bar{\mathbf{k}} \circ \mathbf{R}_i), \ \mathbf{A}' = \text{Re}[\bar{\mathbf{q}}'\bar{\mathbf{k}}'^*].$$

(21)

The rotation matrix (with our 3D hierarchical position embedding) $\mathbf{A}'$ implies relative position in rotation form, and thus boosts the performance.

## A.3  Pose Retrieval Mechanism

For each input sequence, we introduce a learnable token that is initialized within the model and serves as the pose token of the first frame. This token acts as the latent representation of the camera pose at the beginning of the sequence and provides a reference for subsequent frames. In addition to this initialization, we define a compact set of learnable tokens that function as a lightweight pose memory module [63]. This memory is specifically designed for pose retrieval, enabling the model to accumulate and reuse pose-related information across frames in a sequence without incurring excessive computational cost.

After processing each frame, the pose token that has interacted with the current image features is used to update this memory set, thereby maintaining a condensed yet informative representation of previously observed poses. When the next frame arrives, its corresponding pose token is initialized not randomly, but by retrieving an initial value from this pose memory. This retrieval process leverages the image tokens of the current frame to query the memory and select the most relevant pose context. In this way, the proposed memory mechanism allows the model to efficiently capture temporal dependencies between frames while avoiding redundancy and instability.

In the forward pass of $\text{Head}_{global}$, we first generate the pose-modulated tokens using an additional modulation function in CUT3R [63], and then feed them to the DPT architecture to generate the output $\hat{X}_t^{global}$. The modulation function uses two self-attention blocks and modulates the input tokens within the Layer Normalization layers using the pose token.

# B  More Training Details

Table 9 provides detailed information about the datasets used to train our model. To ensure both diversity and robustness, we adopt a multi-stage training strategy that progressively enhances the model's generalization capability across a wide range of real-world and synthetic environments.

| Dataset | Scene Type | Dynamic | Real | Metric |
|---------|-----------|---------|------|--------|
| ARKitScenes [5] | Indoor | Static | Real | Yes |
| BlendedMVS [73] | Mixed | Static | Synthetic | No |
| CO3Dv2 [44] | Object-Centric | Static | Real | No |
| HyperSim [45] | Indoor | Static | Synthetic | Yes |
| MegaDepth [30] | Outdoor | Static | Real | No |
| OmniObject3D [69] | Object-Centric | Static | Synthetic | Yes |
| ScanNet [12] | Indoor | Static | Real | Yes |
| ScanNet++ [74] | Indoor | Static | Real | Yes |
| WildRGBD [70] | Object-Centric | Static | Real | Yes |
| MVS-Synth [24] | Outdoor | Dynamic | Synthetic | Yes |
| PointOdyssey [79] | Mixed | Dynamic | Synthetic | Yes |
| Spring [35] | Mixed | Dynamic | Synthetic | Yes |
| VirtualKITTI2 [7] | Outdoor | Dynamic | Synthetic | Yes |
| Waymo [55] | Outdoor | Dynamic | Real | Yes |

Table 9: **Training Datasets.**

In the first stage, we train our model on a collection of large-scale datasets, including ARK-itScenes, BlendedMVS, CO3Dv2, HyperSim, MegaDepth, ScanNet, ScanNet++, WildRGBD, Vir-tualKITTI2, and Waymo. Such diversity allows the model to learn robust geometric priors and generalizable visual representations that are independent of specific dataset characteristics.

In the second and third stage, we introduce additional datasets and increase the input resolution. This progressive training paradigm enables the model to gradually adapt from coarse-level scene reasoning to high-resolution, detail-oriented prediction, effectively improving its performance in both accuracy and stability.

To further stabilize training in the early phase, we disable the memory fusion mechanism during the first stage. This prevents unstable memory updates that may arise when both pose and feature representations are still under training. Once the model achieves stable convergence, the memory fusion mechanism is introduced to facilitate temporal consistency learning across frames and improve the model's efficiency.

Moreover, we employ a dynamic dataset sampling strategy, where the sampling ratio of each dataset is adaptively adjusted throughout training. This adaptive reweighting ensures that the model receives balanced supervision from different datasets, mitigates potential domain bias, and leads to improved generalization across diverse scene distributions.

## C   More Experiments

**Robustness to Poor Initialization Frames.** The use of the initial frame as the reference leading to possible sensitivity to poor initialization frames is a common issue faced by the "-3R" series of work. We therefore evaluate the robustness of our method on such scenarios. We select Scene Kitchen in the NRGBD dataset and obtain 37 input images by sampling at the interval of 40 frames. To simulate a poor initialization, we set the 23rd sample (corresponding to frame 920 in the original dataset, which mainly contains large areas of the low-texture wall and floor with no other objects) as the first frame and randomly shuffled the order of the remaining frames. The comparison of results between our method and CUT3R under these two input settings is shown in Table 10.

We observe that our method is more robust to the effect of the initialization. Although the performance slightly degrades when the initial frame has significantly degraded content, the drop is acceptable considering the difficulty of such a scenario. We think this is because our explicit pointer memory is aligned with 3D structures of the current scene, which can minimize the impact of initialization as much as possible.

Table 10: **Robustness to poor initialization frames.**

| Method | Normal Initialization | | | | | | Poor Initialization | | | | | |
|---|---|---|---|---|---|---|---|---|---|---|---|---|
| | Acc↓ | | Comp↓ | | NC↑ | | Acc↓ | | Comp↓ | | NC↑ | |
| | Mean | Med. | Mean | Med. | Mean | Med. | Mean | Med. | Mean | Med. | Mean | Med. |
| CUT3R [63] | 0.166 | 0.040 | 0.081 | 0.016 | 0.980 | 0.976 | 0.290 | 0.154 | 0.156 | 0.051 | 0.733 | 0.942 |
| **Ours** | 0.096 | 0.037 | 0.058 | 0.014 | 0.977 | 0.977 | 0.120 | 0.070 | 0.068 | 0.019 | 0.853 | 0.972 |

## D  More Visualizations

We show qualitative results on sparse inputs in Figure 3 in the main text. In this section, we show more qualitative results on dense inputs from static (Figure 5) and dynamic (Figure 6) scenes.

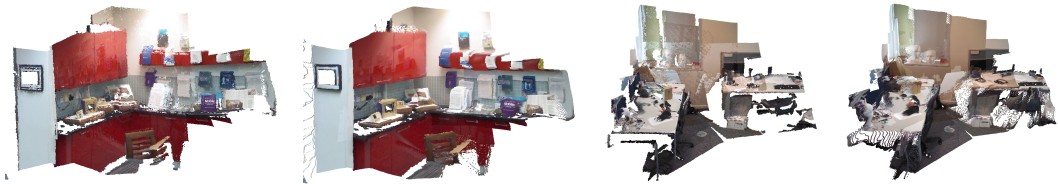

Figure 5: **Qualitative results on dense inputs from static scenes.**

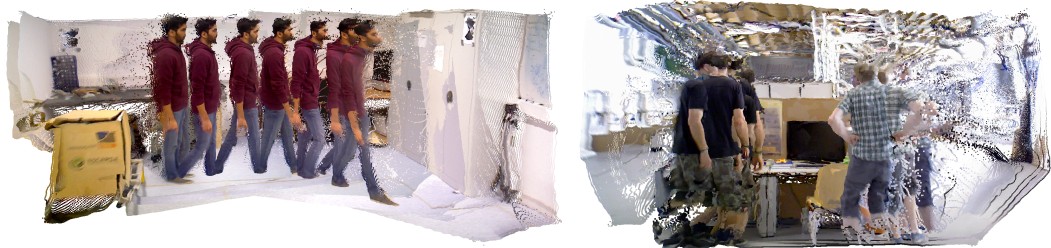

Figure 6: **Qualitative results on dense inputs from dynamic scenes.**

