# OpenReview forum: "Point3R: Streaming 3D Reconstruction with Explicit Spatial Pointer Memory"
_NeurIPS.cc/2025/Conference — NeurIPS 2025 poster_

### Official Review · Reviewer_SeBK · 2025-06-26

**Clarity:** 4
**Significance:** 3
**Originality:** 3
**Rating:** 4
**Confidence:** 5

**Summary:**

In this paper, the authors proposed a framework with an explicit spatial pointer memory, which directly encodes the 3D structure of the reconstructed scene. In this pointer, the authors design a 3D hierarchical position embedding, extend the original RoPE. The authors claimed such integration will be more effective and does not loss any previous state information.

**Questions:**

Please refer to the above reivews.

**Ethical Concerns:**

["NO or VERY MINOR ethics concerns only"]

**Final Justification:**

I kept my initial rating.

**Limitations:**

Please refer to the above reviews.

**Paper Formatting Concerns:**

None.

**Quality:**

4

**Strengths And Weaknesses:**

**Strengths**
1. Such explicit spatial pointer memory is effective and does not lose information theoretically.
2. The extension from original RoPE to 3D is interesting.
3.  The author conducted extensive experiments to support their statements.


**Weaknesses**
1. The architecture is highly similar to Cut3r.
2. Compared to Cut3r, the improvement is not obvious as shown in the presented experiments. And it somehow worse than Cut3r.
3. As stated by the authors, such pointer does not loss any information compared to Cut3r, is more efficient compared to VGGT. I think it is better to conduct more experiments on long sequence so that the comparisons with Cut3r can be more obvious. Besides, I highly suggest the authors to compare with VGGT on the memory usage. I would accept if the authors can make it clear why they did not conduct such comparisons.

---

> ### Author Rebuttal · Authors · 2025-07-31
>
> Thanks for your thoughtful comments, questions and insightful suggestions.
> We provide our detailed responses below.
>
> **1 - About Architectural Similarity with CUT3R**
>
> Our architecture seems similar to CUT3R mainly due to the use of cross-attention for the memory-image interaction, which we think is a general pipeline for information exchange.
> However, for streaming 3D reconstruction methods, we think the core challenge lies in how to design the memory instead of the interaction architecture.
> Existing methods like CUT3R rely on fully implicit feature storage and may suffer from the information loss of previous frames.
> To address this, we propose an explicit pointer memory that directly associates globally aligned information with its real-world locations.
> This dynamic explicit memory design is fundamentally different from that of CUT3R which uses a fixed-size memory.
>
> **2 - Performance Comparison with CUT3R**
>
> We acknowledge that on some specific datasets the improvement over CUT3R is not obvious.
> However, note that our model achieves comparable performance with significantly lower training cost due to the explicit spatial prior.
> Our model was trained for 7 days on 8 GPUs (while CUT3R required around a month) and used fewer than half of the training datasets used in CUT3R.
> This demonstrates the advantage of our proposed explicit pointer memory to enable the model to learn and generalize more effectively.
> We think this training cost reduction is also valuable for promoting research in this field.
>
> Also, our core contribution is to improve the trade-off between information preserving and efficiency for long-term reconstruction with the explicit modeling of memory tokens.
> We verify our advantage on this setting in the next response (**3 - More Comparisons with CUT3R and VGGT**).
>
> Furthermore, we have also observed the following during our research:
>
> - For the data-driven and end-to-end paradigm, more diverse training datasets and more extensive training generally lead to better performance.
> - During training, adjusting the sampling ratio of different types of datasets for each epoch (e.g., increasing the sampling ratio of indoor data while decreasing that of outdoor data) causes the model performance to improve on some while decreasing on others.
>
> Therefore, we think there is still  room for further improvement with larger training costs.
>
> **3 - More Comparisons with CUT3R and VGGT**
>
> In the paper, we use sparsely sampled inputs to evaluate the 3D reconstruction performance to demonstrate that our explicit pointer memory (associated with 3D spatial locations) is effective and does not rely on similarity or continuity between input frames.
> However, we agree that your suggestion to evaluate on longer sequences is very reasonable.
> Therefore, we provide comparisons on longer sequences sampled from the 7-scenes and NRGBD datasets.
> Specifically, for each scene in 7-scenes, we sample a sequence with a frame interval of 10 (each sequence contains 50-100 frames).
> For each scene in NRGBD, we sample a sequence with a frame interval of 20 (each sequence contains 40-80 frames).
> As shown in the table below, the results demonstrate the superiority of our method in long-sequence reconstruction, which benefits from the explicit pointer memory that effectively stores information from past frames.
>
> | Method | Dataset | Acc (mean) ↓ | Acc (med) ↓ | Comp (mean) ↓ | Comp (med) ↓ | NC (mean) ↑ | NC (med) ↑ |
> |:------:|:-------:|:-----:|:-----:|:-----:|:-----:|:-------:|:--------:|
> | CUT3R  | 7-scenes | 0.038 | 0.019 | 0.029 | 0.008 | 0.623   | 0.693    |
> | **Ours**   | 7-scenes | **0.035** | **0.017** | **0.024** | **0.007** | **0.627**   | **0.701**    |
> | CUT3R  | NRGBD   | 0.093 | 0.043 | 0.037 | 0.010 | 0.744   | 0.906    |
> | **Ours**   | NRGBD   | **0.073** | **0.034** | **0.026** | **0.008** | **0.758**   | **0.917**    |
>
> As noted by the official NeurIPS guidelines ("papers that appeared online after March 1st, 2025 will generally be considered 'contemporaneous'"), we considered VGGT as a contemporaneous work (public on arXiv on March 14th) and therefore only discussed but not compared intensely in the paper.
> Our model targets the streaming frame-by-frame online reconstruction setting while VGGT targets one-shot global reconstruction with all inputs, which usually causes a significant drop on the runtime efficiency.
> We use the following table to compare the peak runtime memory usage of our method and VGGT under different numbers of input frames.
>
> | Method\Frames | 5 | 10 | 20 | 30 | 40 |
> |:------:|:-------:|:-----:|:-----:|:-----:|:-----:|
> | VGGT | 3.74 GB | 5.59 GB | 7.51 GB | 9.43 GB | 11.44 GB |
> | Ours | 3.60 GB | 3.69 GB | 4.02 GB | 4.47 GB | 4.93 GB |
>
> We hope the above response can help address your concerns. We are happy to answer any additional questions you may have.

---

> > ### Comment · Reviewer_SeBK · 2025-08-04
> >
> > Thanks to the authors for the rebuttal. Some of my concerns are addressed. However, I still have some concerns.
> >
> > Firstly, I understand that the authors used fewer datasets and training time to train their model; however, the authors' observation cannot convince me, since such observations are not supported by any experiments. Besides, I think more training data does not mean better performance, since the performance can also be limited by the model architecture, such as how many learnable parameters and blocks within the model. Secondly, it seems the authors did not conduct the experiments on the whole sequences of longer videos. Why? I would like to know the exact frame count of each sampled sequence. What I mean by long videos is like 500-1000 frames. I am looking forward to more discussion. Thank you.

---

> > > ### Author Response · Authors · 2025-08-06
> > >
> > > We thank the reviewer for the timely reply and constructive suggestions, which are greatly helpful in improving our paper. We appreciate the opportunity to further clarify your concerns.
> > >
> > > **1- Effect of extended training on our model performance**
> > >
> > > Sorry for the confusion. We agree that more training data and iterations do not necessarily lead to better performance due to the model capacity. We meant that for our model with ~1B parameters, training for a longer time could potentially improve the performance.
> > > To verify this, we compare the performance of our model trained with different durations.
> > > As shown in the table below, compared to the version reported in the paper trained for 7 days, our model achieves a better performance after 20 days of training.
> > >
> > > | Method | Dataset | Acc (mean) ↓ | Acc (med) ↓ | Comp (mean) ↓ | Comp (med) ↓ | NC (mean) ↑ | NC (med) ↑ |
> > > |:------:|:-------:|:-----:|:-----:|:-----:|:-----:|:-------:|:--------:|
> > > | CUT3R (30 days)  | 7-scenes | 0.126 | 0.047 | 0.154 | 0.031 | 0.727   | 0.834    |
> > > | Ours (7 days)   | 7-scenes | 0.124 | 0.058 | 0.139 | 0.054 | 0.725   | 0.834    |
> > > | Ours (20 days)   | 7-scenes | **0.085** | **0.046** | **0.087** | **0.030** | **0.739**   | **0.854**    |
> > > | CUT3R (30 days)  | NRGBD | 0.099 | 0.031 | 0.076 | **0.026** | **0.837**   | **0.971** |
> > > | Ours (7 days)   | NRGBD | 0.079 | 0.031 | 0.073 | 0.027 | 0.824   | 0.965    |
> > > | Ours (20 days)   | NRGBD | **0.077** | **0.030** | **0.069** | 0.027 | 0.835   | **0.971**  |
> > >
> > > **2- More evaluation on very long sequences**
> > >
> > > Thanks for the suggestion for further evaluation on the whole sequences of longer videos (500-1000 frames).
> > > We resampled the testing sequences from 7-scenes and NRGBD, and below are the exact frame count of each sampled sequence.
> > >
> > > |7-scenes|pumpkin-01|pumpkin-07|office-02|office-06|office-07|office-09|fire-03|fire-04|redkitchen-03|
> > > |:-------:|:-----:|:-----:|:-----:|:-----:|:-----:|:-----:|:-----:|:-----:|:-----:|
> > > |Length of the whole sequences|1000|1000|1000|1000|1000|1000|1000|1000|1000|
> > > |Sampled frame count|1000|1000|1000|1000|1000|1000|1000|1000|1000|
> > > |7-scenes|redkitchen-04|redkitchen-06|redkitchen-12|redkitchen-14|stairs-01|stairs-04|heads-01|chess-03|chess-05|
> > > |Length of the whole sequences|1000|1000|1000|1000|500|500|1000|1000|1000|
> > > |Sampled frame count|1000|1000|1000|1000|500|500|1000|1000|1000|
> > > |NRGBD|whiteroom|grey_white_room|morning_apartment|green_room|staircase|complete_kitchen|breakfast_room|kitchen|thin_geometry|
> > > |Length of the whole sequences|1676|1493|920|1442|1149|1211|1167|1517|395|
> > > |Sampled frame count|838|747|920|721|575|606|584|759|395|
> > >
> > > We compared our method with CUT3R using the long sequences above.
> > > As shown in the table below, our method outperforms CUT3R for a large margin, showing the advantage of our model to effectively store more information from past frames on long sequences.
> > >
> > > | Method | Dataset | Acc (mean) ↓ | Acc (med) ↓ | Comp (mean) ↓ | Comp (med) ↓ | NC (mean) ↑ | NC (med) ↑ |
> > > |:------:|:-------:|:-----:|:-----:|:-----:|:-----:|:-------:|:--------:|
> > > | CUT3R  | 7-scenes | 0.238 | 0.172 | 0.105 | 0.025 | 0.527   | 0.537    |
> > > | **Ours**   | 7-scenes | **0.071** | **0.033** | **0.031** | **0.015** | **0.558**| **0.587** |
> > > | CUT3R  | NRGBD   | 0.372 | 0.270 | 0.211 | 0.090 | 0.556   |0.582|
> > > | **Ours**   | NRGBD   | **0.110** | **0.050** | **0.025** | **0.009** | **0.641**   | **0.729** |
> > >
> > >
> > > We hope the above response can help address your concerns. We are happy to answer any additional questions you may have.

---

> > > > ### Comment · Reviewer_SeBK · 2025-08-06
> > > >
> > > > Thanks to the authors for further clarification. My concerns are addressed. Will check the discussions between the authors and other reviewers, and modify my rating accordingly. Currently, I am inclined to keep my positive rating. Thank you.

---

### Official Review · Reviewer_LEkT · 2025-06-30

**Clarity:** 1
**Significance:** 2
**Originality:** 3
**Rating:** 4
**Confidence:** 3

**Summary:**

This paper proposes an end-to-end feed-forward structure-from-motion method to sequentially register cameras and estimate scene structure by incorporating features to each estimated 3D points.
Based on the Euclidean distance between 3D points estimated by different images and dynamic threshold, previously estimated positions and features of 3D points are updated by the latest registered image.
By using hierarchical position embedding, the proposed decoder injects relative position information into image tokens and memory features.

**Questions:**

1. For training objective in Eq.(17)(18)，how to use the train data without metric? Directly using the scale from the ground truth may make the model learn the wrong metric.
2. Where is $z_t$ from ?Is there any relationship between $z_t$, $z_{t-1}$, and $z_0$?
3. Point3R is an incremental method where cameras are registered sequentially. On the one hand, the accuracy of the newly registered camera poses depends on the estimated 3D points, which may suffer from error accumulation in large scenes with long camera trajectories. How can the error accumulation problem be solved in Point3R without global alignment?
On the other hand, does the order of camera registration impact the reconstruction result? This aspect should be discussed in detail.
On the other hand, whether the order of registering cameras impacts the reconstruction result, which should be discussed in detail.
4. For the position embedding of image tokens with unordered inputs, why does the incorrect position embedding not introduce adverse effects? Is this embedding method sensitive to the order of image registration?
5. What are the advantages of Point3R compared to VGGT in efficiency and scalabily?Since all the 3D points are reused for registering new images, this also severely impacts efficiency and scalability. Moreover, for online applications such as embodied agents, what are the advantages of Point3R compared to the dynamic SLAM method MegaSAM?

**Ethical Concerns:**

["NO or VERY MINOR ethics concerns only"]

**Final Justification:**

Thank you for the author's reply. Most of my doubts have been resolved, and I maintain my original rating.

**Limitations:**

yes

**Quality:**

3

**Strengths And Weaknesses:**

**Strengths:**
By integrating explicit positions and features of 3D points, this paper introduces an innovative end-to-end incremental structure-from-motion method that demonstrates better scalability than global end-to-end methods, such as VGGT, and avoids the need to estimate pairs of input images, as in Dust3r. By using the 3D position embedding, the decoder of Point3R can leverage the relative positional information of 3D points for pointer-image interaction.

**Weaknesses:**
The accuracy of camera poses is crucial for 3D reconstruction, as the scene structure can be further refined with fixed camera poses. However, compared to methods using global optimization, the proposed Point3R does not show improved accuracy in camera poses. From a practical perspective, the advantages over existing methods remain unclear.

---

> ### Author Rebuttal · Authors · 2025-07-31
>
> Thanks for your thoughtful comments, questions and insightful suggestions.
> We provide detailed responses below.
>
> **W1 - Accuracy of camera poses**
>
> Thanks for the question.
> We agree that global optimization methods generally produce more accurate pose estimation.
> However, they usually require much more time for optimization, and the motivation of memory-based streaming methods is to eliminate the dependency on global optimization to improve the runtime efficiency.
> Optionally, we can add a global optimization step as post-processing to obtain more accurate pose estimation when time permits.
>
> We further compare the runtime efficiency between our method and DUSt3R (with global optimization) in the table below.
>
> | Method | #Frames | Runtime (s) | #Frames | Runtime (s) | #Frames | Runtime (s) |
> |:------:|:-------:|:-----:|:-----:|:-------:|:-----:|:-----:|
> | DUSt3R | 10 | 9.76 | 20 | 38.55 | 30 | 88.67  |
> | Ours | 10 | 2.03 | 20 | 4.10 | 30 | 6.42 |
>
> This verifies the advantage of our method in efficiency.
>
> We will address your concerns more thoroughly using the answers given below (**About the error accumulation problem and the impact of the camera orders on the reconstruction quality**, **Comparison with VGGT and MegaSAM**).
>
> **Q1 - About training data without metric**
>
> Sorry for the confusion.
> As stated in Line 197-198, we only use the scale from the ground truth when the data is metric.
> For training data without metric, we apply their own scale normalization factors for the groundtruth and the predictions accordingly.
> In this no-metric setting, we expect the model to learn a geometry distribution which is invariant to scale, thereby mitigating the influence of potentially inaccurate or inconsistent annotated scales with respect to the real world.
>
> **Q2 - More details about the pose token $z$**
>
> Sorry for the confusion.
> For each input sequence, we use a learnable token (which is initialized within the model) to serve as the pose token $z_{0}$ of the first frame.
> We also initialize a set of learnable tokens to serve as a lightweight memory module specifically designed for pose retrieval.
> After processing each frame, we update this set of tokens (the "pose memory") using the pose token $z_{t}^{'}$ of the current frame which has participated in interaction.
> When adding the pose token for the next frame, we leverage its image tokens to "retrieve" an initial value for this frame’s pose token from this "pose memory".
> In other words, the pose tokens $z_{0}, ..., z_{t-1}, z_{t}$ of different frames are connected through this shared token set.
> We apologize for the confusion and will include these details in the revised version.
>
> **Q3 - About the error accumulation problem and the impact of the camera orders on the reconstruction quality**
>
> Thanks for the valuable question.
> Streaming methods targeting real-time incremental reconstruction inevitably suffer from the problem of error accumulation, especially in large scenes with long camera trajectories.
> The mitigation of this issue motivates us to store past information in the 3D space which the model can directly access when processing the current frame to reduce the influence of frame-to-frame cascading dependencies.
>
> To verify this, the table below compares our performance with CUT3R under a long-sequence reconstruction setting.
> Specifically, for each scene in 7-scenes, we sample a sequence with a frame interval of 10 (each sequence contains 50-100 frames).
> For each scene in NRGBD, we sample a sequence with a frame interval of 20 (each sequence contains 40-80 frames).
>
> | Method | Dataset | Acc (mean) ↓ | Acc (med) ↓ | Comp (mean) ↓ | Comp (med) ↓ | NC (mean) ↑ | NC (med) ↑ |
> |:------:|:-------:|:-----:|:-----:|:-----:|:-----:|:-------:|:--------:|
> | CUT3R  | 7-scenes | 0.038 | 0.019 | 0.029 | 0.008 | 0.623   | 0.693    |
> | **Ours**   | 7-scenes | **0.035** | **0.017** | **0.024** | **0.007** | **0.627**   | **0.701**    |
> | CUT3R  | NRGBD   | 0.093 | 0.043 | 0.037 | 0.010 | 0.744   | 0.906    |
> | **Ours**   | NRGBD   | **0.073** | **0.034** | **0.026** | **0.008** | **0.758**   | **0.917**    |
>
> This validates that explicit memory storage alleviates the error accumulation issue.
>
> Our method is also compatible with global alignment as a post-processing step to further reduce the impact of error accumulation when necessary.
> However, since this would significantly compromise runtime efficiency, we still target the feedforward end-to-end streaming reconstruction without global alignment primarily.
>
> We also evaluate the robustness of our model to the order of camera registration.
> We observe from the table below that our method still achieves good reconstruction performance from shuffled inputs.
> To be specific, for each scene in 7-scenes, we sample a sequence with a frame interval of 20 (each sequence contains 25-50 frames).
> For each scene in NRGBD, we sample a sequence with a frame interval of 40 (each sequence contains 20-40 frames).
> Then we disrupt the input sequence after the sampling to evaluate the reconstruction results.
>
> | Sample | Dataset | Acc (mean) ↓ | Acc (med) ↓ | Comp (mean) ↓ | Comp (med) ↓ | NC (mean) ↑ | NC (med) ↑ |
> |:------:|:-------:|:-----:|:-----:|:-----:|:-----:|:-------:|:--------:|
> | Ordered  | 7-scenes | 0.032 | 0.016 | 0.024 | 0.008 | 0.665   | 0.758    |
> | Shuffled   | 7-scenes | 0.033 | 0.014 | 0.019 | 0.010 | 0.669   | 0.764    |
> | Ordered  | NRGBD   | 0.064 | 0.031 | 0.029 | 0.011 | 0.801   | 0.949    |
> | Shuffled   | NRGBD   | 0.063 | 0.027 | 0.029 | 0.009 | 0.800   | 0.953    |
>
>
> **Q4 - About the 3D position embedding**
>
> Our method is mainly designed for continuous exploration and streaming reconstruction in real-world application scenarios, and there is often a certain continuity between the streaming inputs.
> Therefore, we use this 3D position embedding to serve as spatial priors.
> However, when there is a significant shift between the current and previous frames, we expect the data-driven training to teach the model to correct the potential deviations of this spatial prior.
> This is supported by the reconstruction results with unordered inputs discussed in the previous response (**About the error accumulation problem and the impact of the camera orders on the reconstruction quality**), and also the reconstruction results shown in our paper where the input frames have low spatial overlap.
>
> **Q5 - Comparison with VGGT and MegaSAM**
>
> On the streaming setting, VGGT needs to use all the past frames and conducts the all-to-all attention to register the latest frame, which causes a significant drop in runtime efficiency.
> The following table compares the per-frame runtime and peak memory usage of our method and VGGT.
>
> | Method | #Frames | Runtime (per frame) | Memory Usage (peak) | #Frames | Runtime (per frame) | Memory Usage (peak) | #Frames | Runtime (per frame) | Memory Usage (peak) |
> |:------:|:-------:|:-----:|:-----:|:-------:|:-----:|:-----:|:-------:|:-----:|:-----:|
> | VGGT | 10 | 0.37 s | 5.59 GB | 20 | 0.87 s | 7.51 GB | 30 | 1.32 s | 9.43 GB |
> | Ours | 10 | 0.20 s | 3.69 GB | 20 | 0.20 s | 4.02 GB | 30 | 0.21 s | 4.47 GB |
>
> We do not use all the 3D points to register the new inputs.
> To prevent an unreasonable surge in the pointer count as the number of frames increases, we dynamically adjust the threshold used for nearest-neighbor search during the memory fusion module based on the spatial extent of the currently reconstructed 3D scene (details of this threshold computation are provided in the supplementary material).
> Based on our statistics across all test sequences on 7-scenes and NRGBD datasets (each sequence contains 50–100 frames), this dynamic threshold adjustment leads to a stabilization of the total number of pointers at around 1000–2000 after a certain number of frames.
> Consequently, the per-frame processing time also stabilizes at approximately 0.2 seconds (as shown in Figure 4 in the main paper).
>
> Dynamic SLAM methods like MegaSAM accumulate a sufficient number of keyframes to initialize the exploring system, and often require external monocular depth priors.
> They can achieve more consistent depth estimation across the sequence after obtaining calibrated poses.
> In contrast, our method is a more fundamental end-to-end model that jointly estimates dense geometry and camera pose for each frame using only past observations, without requiring global optimization or full-sequence access.
> This makes it more practical and flexible for real-time applications.
> Importantly, our design remains compatible with global alignment techniques if needed, enabling further improvement in accuracy without sacrificing online performance.
>
> We hope the above response can help address your concerns. We are happy to answer any additional questions you may have.

---

> ### Comment · Reviewer_LEkT · 2025-08-07
>
> Appreciated for the feedback of the authors. For streaming 3D reconstruction applications, such as embodied robotics, it remains unclear whether Point3R demonstrates a clear advantage over MegaSAM in terms of camera pose accuracy. In particular, all supplementary experiments were conducted on fewer than 50 images, which is insufficient to convincingly demonstrate the method’s superiority in practical applications. Additionally, error accumulation cannot be effectively mitigated simply by using explicit 3D points, as they are also subject to drift.

---

> > ### Author Response · Authors · 2025-08-08
> > **Comment-Part 1**
> >
> > We thank the reviewer for the timely reply and constructive suggestions, which are greatly helpful in improving our paper.
> > We appreciate the opportunity to further clarify your concerns.
> >
> > **1-Detailed comparison with MegaSAM in terms of camera pose estimation**
> >
> > We apologize for the confusion.
> >
> > We also think MegaSAM is a very good solution to obtain relatively accurate camera pose estimation from videos.
> > It follows a typical SLAM paradigm, where the main objective lies in accurately registering the camera poses from RGB-D inputs (where the depth could be obtained from prediction models).
> > Differently, feed-forward 3D reconstruction methods (e.g., DUSt3R, CUT3R, and ours) aim to develop a unified model to predict 3D reconstructed points for each frame in a shared coordinate system (and some recent methods simultaneously predict the camera pose with an additional head).
> >
> > More specifically, MegaSAM relies on existing depth estimation models (for per-frame depth and focal length estimation) and cannot independently produce camera pose estimates from real-time streaming inputs.
> > Differently, our method can produce both dense reconstructed points and camera poses with per-frame streaming RGB input.
> >
> > Therefore, we would like to clarify that pose estimation is only one task of our model and is not the core objective.
> > Still, following the suggestion of the reviewer, we use the following table to further explain the difference between MegaSAM and our method on the camera pose estimation task.
> >
> > |Method|Pipeline Overview|Advantages|Disadvantages|
> > |:-|:-|:-|:-|
> > |MegaSAM|Use DepthAnything and UniDepth to obtain per-frame depth and the focal length → Select keyframes to register the poses and perform local BA in a sliding window manner → Register the poses of other frames and conduct the global BA|Higher estimation accuracy|Complex pipeline; Dependency on DepthAnything and UniDepth models|
> > |Ours|Directly process the visual inputs frame by frame and output the corresponding poses|End-to-end paradigm; No dependency on any other modules|Still has room for improvement in estimation accuracy (without post process)|
> >
> > We thus think directly comparing the pose estimation accuracy between these two fundamentally different paradigms is not fair.
> > Still, we agree that this comparison would be meaningful and beneficial to the community to analyze the pros and cons of the two paradigms.
> > We therefore follow the reviewer's suggestion and provide comparisons on the setting of MegaSAM.
> > For fairness to some extent, we extend CUT3R and our method with a global bundle adjustment module to estimate the poses on the Sintel dataset.
> >
> > |Method|Type|ATE|RTE|RRE|
> > |:-:|:-:|:-:|:-:|:-:|
> > |CasualSAM|SLAM and SfM|0.067|0.019|0.47|
> > |ACE-Zero|SLAM and SfM|0.065|0.028|1.92|
> > |Particle-SfM|SLAM and SfM|0.057|0.038|1.64|
> > |**MegaSAM**|SLAM and SfM|0.023|0.008|0.06|
> > |MonST3R|Feed-forward Reconstrcution|0.078|0.038|0.49|
> > |CUT3R|Feed-forward Reconstrcution|0.031|0.016|0.33|
> > |**Ours**|Feed-forward Reconstrcution|0.023|0.012|0.14|
> >
> > Again, we would like to clarify that it is not our objective to outperform MegaSAM on the pose estimation task. This is why we mainly conducted experiments following the setting of the other 3D reconstruction methods (e.g., CUT3R). We believe those can provide a better evaluation of our model.
> >
> > Please check Comment-Part 2 for more responses. Thank you.

---

> > ### Author Response · Authors · 2025-08-08
> > **Comment-Part 2**
> >
> > **2-More evaluation on very long sequences**
> >
> > Thanks for this suggestion.
> > To better demonstrate our method's superiority in practical applications, we resampled the testing sequences from 7-scenes at the interval of 1 (each sequence contains 500-1000 frames) and NRGBDs at the interval of 2 (each sequence contains 400-900 frames).
> > We compared our method with CUT3R using the long sequences above.
> > As shown in the table below, our method outperforms CUT3R by a large margin, showing the advantage of our model when handling long sequence input in practical applications.
> > | Method | Dataset | Acc (mean) ↓ | Acc (med) ↓ | Comp (mean) ↓ | Comp (med) ↓ | NC (mean) ↑ | NC (med) ↑ |
> > |:-:|:-:|:-:|:-:|:-:|:-:|:-:|:-:|
> > | CUT3R  | 7-scenes | 0.238 | 0.172 | 0.105 | 0.025 | 0.527   | 0.537    |
> > | **Ours**   | 7-scenes | **0.071** | **0.033** | **0.031** | **0.015** | **0.558**| **0.587** |
> > | CUT3R  | NRGBD   | 0.372 | 0.270 | 0.211 | 0.090 | 0.556   |0.582|
> > | **Ours**   | NRGBD   | **0.110** | **0.050** | **0.025** | **0.009** | **0.641**   | **0.729** |
> >
> > **3-Error accumulation problem**
> >
> > Thanks for the comment and the opportunity to clarify.
> > We mitigate the error accumulation by extending the memory to try to preserve more direct information.
> > However, directly preserving all the past tokens would cause a very heavy computation burden over time, and there is then a trade-off between information preserving and efficiency.
> > The objective of using explicit 3D points is then to improve this trade-off by merging tokens that are close in the 3D space.
> > This can keep the total number of memory tokens within a reasonable range while covering the observed area.
> >
> > We update the pointer memory to incorporate the newest information.
> > Note that some memories are not updated (unobserved by the current frame) and can be directly accessed to mitigate the error accumulation.
> >
> > To verify this, we think the results in the table from the second answer (**More evaluation on very long sequences**) help to demonstrate the significant improvement achieved by our method when conducting long sequence reconstruction.
> >
> > We hope the above response can help address your concerns. We are happy to answer any additional questions you may have.

---

> ### Comment · Reviewer_LEkT · 2025-08-08
>
> Thank you for the author's reply. Most of my doubts have been resolved, and I maintain my original rating.

---

### Official Review · Reviewer_JUWw · 2025-07-01

**Clarity:** 3
**Significance:** 3
**Originality:** 4
**Rating:** 5
**Confidence:** 5

**Summary:**

Point3R is an online framework for dense streaming 3D reconstruction from image sequences or collections. It addresses the limitations of existing methods, such as limited implicit memory capacity and information loss, by maintaining an explicit spatial pointer memory directly associated with the current scene's 3D structure. The framework also incorporates a 3D hierarchical position embedding to facilitate interaction between the latest frame and the pointer memory, and a simple yet effective fusion mechanism to ensure memory uniformity and efficiency. Point3R achieves competitive performance in various 3D/4D tasks, including dense 3D reconstruction, monocular and video depth estimation, and camera pose estimation, all while maintaining low training costs.

**Questions:**

1.Robustness to Poor Initialization Frames
How does the system perform in terms of overall reconstruction quality when the first frame has significantly degraded content (e.g., facing the floor or a low-texture corner of a room)? Since the initial frame often serves as a reference for pose estimation and scene initialization, does this lead to cascading errors in reconstruction?

2.Explicit vs. Implicit Spatial Memory for Challenging Cases
Would an explicit spatial memory mechanism (e.g., voxel grids or feature maps) provide more reliable handling of such edge cases compared to the implicit memory representations used in CUT3R/Spann3R? Explicit representations might offer better stability for poorly initialized scenes, but could this come at the cost of flexibility or scalability?

**Ethical Concerns:**

["NO or VERY MINOR ethics concerns only"]

**Final Justification:**

All my concerns are addressed.

**Limitations:**

See weakness.

**Paper Formatting Concerns:**

No issue.

**Quality:**

4

**Strengths And Weaknesses:**

Strengths:\
1.This paper presents a clear and well-structured contribution to streaming 3D reconstruction through its novel Point3R framework. The writing flows logically from problem formulation to technical solutions, with key concepts like the spatial pointer memory and 3D hierarchical position embedding introduced intuitively before detailed explanations. Visual elements like comparative figures and results tables effectively support the textual content, while mathematical formulations are properly derived and referenced.

2.The technical approach demonstrates both novelty and soundness through its explicit spatial pointer memory design, which addresses critical limitations of prior implicit memory methods. By anchoring memory units to 3D positions and incorporating a biologically-inspired structure, the method avoids redundancy and information loss while enabling dynamic updates. The innovative 3D hierarchical position embedding extends existing techniques to better handle spatial relationships, and the memory fusion mechanism ensures efficient scaling. Comprehensive experiments across multiple tasks and datasets validate the framework's robustness.

3.The work is well-motivated by clearly identifying gaps in current approaches, particularly the inefficiencies of pair-wise methods and limitations of existing memory architectures. The biological analogy to human spatial memory provides a strong conceptual foundation, while the practical focus on streaming applications aligns with real-world needs in robotics and AR/VR. Despite building on some existing components, the core innovations and their empirical validation demonstrate significant advancement over prior work.

Weakness:\
1.Computational Efficiency Degradation Over Time:
As the video sequence progresses, the computational processing speed per frame gradually decreases. This performance degradation becomes particularly problematic for long video sequences, potentially leading to suboptimal results in later frames. Are there any potential solutions or optimizations to mitigate this temporal efficiency decay?

2.Absence of Visual Demonstration for Unordered Input:
The paper currently lacks visual examples or experimental results demonstrating the method's performance when handling unordered input sequences. Including such illustrations would better showcase the algorithm's capability to process temporally inconsistent or randomly arranged frames.

3.Suboptimal Performance on Certain Benchmarks:
While the method shows competitive results overall, its performance does not achieve state-of-the-art (SOTA) levels on some specific datasets. This suggests there may be room for improvement in adapting the approach to different data characteristics or scenarios.

---

> ### Author Rebuttal · Authors · 2025-07-31
>
> Thanks for your thoughtful comments, questions and insightful suggestions.
> We provide our detailed responses below.
>
> **1 - Computational Efficiency Degradation over Time**
>
> Thanks for the question.
> There is a trade-off between long-term information preserving and efficiency, and our core motivation is to improve this trade-off.
> To preserve more information, a straightforward solution is to keep all past memories instead of replacing them.
> However, as the number of frames increases, such storage becomes undoubtedly redundant and inefficient.
> This motivates us to explore how to effectively fuse these redundant tokens, and we achieve this by fusing tokens that are close in the 3D space.
>
> Specifically, we dynamically adjust the threshold used for nearest-neighbor search during the memory fusion module, based on the spatial extent of the currently reconstructed 3D scene.
> Based on our statistics across all test sequences on 7-scenes and NRGBD datasets (each sequence contains 50–100 frames), this dynamic threshold adjustment leads to a stabilization of the total number of pointers at around 1000–2000 after a certain number of frames.
> Consequently, the per-frame processing time also stabilizes at approximately 0.2 seconds (as shown in Figure 4 in the original paper).
>
> Furthermore, as a promising future direction, we believe that introducing a sliding window mechanism to selectively filter the memory pointers involved in interaction could potentially further improve this efficiency.
> During the real-world long video captures, the newly captured regions are usually adjacent to those of the most recent frames.
> Therefore, only using the pointers stored from a number of recent frames may be sufficient for effective and efficient interaction.
>
> **2 - More Experimental Results for Unordered Input**
>
> Thanks for the suggestion.
> We think that in real-world application scenarios, online exploration and reconstruction of the current environment exhibit a certain degree of continuity and thus did not include results based on unordered inputs.
> Still, we agree that evaluating 3D reconstruction performance under unordered inputs is important for demonstrating the robustness of our method.
> According to the guidelines, we cannot include images to present the visual results (which we will add to the paper) in the rebuttal, we therefore only provide the quantitative results in the table below.
> Specifically, for each scene in 7-scenes, we sample a sequence with a frame interval of 20 (each sequence contains 25-50 frames).
> For each scene in NRGBD, we sample a sequence with a frame interval of 40 (each sequence contains 20-40 frames).
>
> | Sample | Dataset | Acc (mean) ↓ | Acc (med) ↓ | Comp (mean) ↓ | Comp (med) ↓ | NC (mean) ↑ | NC (med) ↑ |
> |:------:|:-------:|:-----:|:-----:|:-----:|:-----:|:-------:|:--------:|
> | Ordered  | 7-scenes | 0.032 | 0.016 | 0.024 | 0.008 | 0.665   | 0.758    |
> | Shuffled | 7-scenes | 0.033 | 0.014 | 0.019 | 0.010 | 0.669   | 0.764    |
> | Ordered  | NRGBD   | 0.064 | 0.031 | 0.029 | 0.011 | 0.801   | 0.949    |
> | Shuffled | NRGBD   | 0.063 | 0.027 | 0.029 | 0.009 | 0.800   | 0.953    |
>
> We observe that our method still achieves good reconstruction performance after the input sequence is shuffled.
> This verifies the robustness of our explicit pointer memory to handle the discontinuity of the inputs.
>
> **3 - Suboptimal Performance on Certain Benchmarks**
>
> We acknowledge that on some specific datasets our method does not achieve state-of-the-art and there is room for improvement.
> We have observed the following during our research:
> - For the data-driven and end-to-end paradigm, more diverse training datasets and more extensive training generally lead to better performance.
> - During training, adjusting the sampling ratio of different types of datasets for each epoch (e.g., increasing the sampling ratio of indoor data while decreasing that of outdoor data) causes the model performance to improve on some while decreasing on others.
>
> Also note that our model achieves comparable performance with significantly lower training cost due to the explicit spatial prior.
> Our model was trained for 7 days on 8 GPUs (while CUT3R required around a month) and used fewer than half of the training datasets used in CUT3R.
> This demonstrates the advantage of our proposed explicit pointer memory to enable the model to learn and generalize more effectively, which still has room for further improvement with larger training costs.
> We think this training cost reduction is also valuable for promoting research in this field.
>
>
> **4 - Robustness to Poor Initialization Frames**
>
> Thanks for the suggestion.
> The use of the initial frame as the reference leading to possible sensitivity to poor initialization frames is a common issue faced by the "-3R" series of work.
> We therefore evaluate the robustness of our method on such scenarios.
> We select Scene Kitchen in the NRGBD dataset and obtain 37 input images by sampling at the interval of 40 frames.
> To simulate a poor initialization, we set the 23rd sample (corresponding to frame 920 in the original dataset, which mainly contains large areas of the low-texture wall and floor with no other objects) as the first frame and randomly shuffled the order of the remaining frames.
> The comparison of results between our method and CUT3R under these two input settings is shown in the table below.
>
> | Method | Initialization | Acc (mean) ↓ | Acc (med) ↓ | Comp (mean) ↓ | Comp (med) ↓ | NC (mean) ↑ | NC (med) ↑ |
> |:------:|:-------:|:-----:|:-----:|:-----:|:-----:|:-------:|:--------:|
> | CUT3R  | Normal | 0.166 | 0.040 | 0.081 | 0.016 | 0.980   | 0.976    |
> | CUT3R   | Poor | 0.290 | 0.154 | 0.156 | 0.051 | 0.733   | 0.942    |
> | Ours  | Normal   | 0.096 | 0.037 | 0.058 | 0.014 | 0.977   | 0.977    |
> | Ours   | Poor   | 0.120 | 0.070 | 0.068 | 0.019 | 0.853   | 0.972    |
>
> We observe that our method is more robust to the effect of the initialization.
> Although the performance slightly degrades when the initial frame has significantly degraded content, the drop is acceptable considering the difficulty of such a scenario.
> We think this is because our explicit pointer memory is aligned with 3D structures of the current scene, which can minimize the impact of initialization as much as possible.
>
> **5 - Explicit vs. Implicit Spatial Memory for Challenging Cases**
>
> From the experimental comparisons provided in the previous response, it can be observed that our explicit pointer memory is more robust than implicit memory representations when faced with poor scene initialization.
> Still, we think our pointer memory is also flexible and scalable to a certain extent.
> Each pointer is directly associated with a 3D position in the scene, and this position is not fixed.
> Also, as mentioned in the answer to the first question (**Computational Efficiency Degradation over Time**), the number of memory pointers is directly related to the structural complexity of the current scene.
> The ability to dynamically and flexibly adjust the memory results in that our method relying less on the specific scene initialization and reaching a stabilization of the total number of memory tokens, thus offering high flexibility and scalability.
>
> We hope the above response can help address your concerns. We are happy to answer any additional questions you may have.

---

> > ### Comment · Reviewer_JUWw · 2025-08-04
> >
> > Thank you for the detailed response. To follow up, could you please explain the key differences between implicit and explicit memory within the context of Embodied AI and large language models?
> >
> > Specifically, please elaborate on:
> >
> > 1. How do these two memory types affect model design and training?
> >
> > 2. What are the primary advantages and disadvantages of each?
> >
> > 3. For what specific scenarios or use cases is each type of memory best suited?

---

> > > ### Author Response · Authors · 2025-08-05
> > >
> > > We thank the reviewer for the timely reply and constructive suggestions, which are greatly helpful in improving our paper.
> > >
> > > Implicit memory is usually represented by cached tokens or features. For embodied AI, explicit memory usually denotes spatial-aware representations in the 3D space. For large language models, explicit memory usually denotes additional texts (e.g., in external documents).
> > >
> > > **1- How do these two memory types affect model design and training?**
> > >
> > > Implicit memory caches all past implicit tokens and directly uses these tokens to interact with the latest information, which usually advocates a transformer architecture.
> > > The core of the explicit memory model design lies in how to organize the explicit representations in a compact and efficient manner, and how to enable effective interaction (e.g., through positional embedding) between the explicit memory and the latest information.
> > > Models with implicit memory usually rely on large-scale training data to be effectively optimized.
> > > In contrast, explicit memory offers direct, efficient, and accessible storage of past information for each sample, which lowers the overall demand for large-scale training data.
> > >
> > > **2- What are the primary advantages and disadvantages of each?**
> > >
> > > We use the following table to compare the main advantages and disadvantages of these two types of memory.
> > >
> > >
> > > | Method |   Advantage | Disadvantage |
> > > |:------|:-------|:-----|
> > > | Implicit memory | Simpler model architecture | Poor capability in retaining long-term information; Heavy reliance on large and diverse datasets for training and generalization |
> > > | Explicit memory | Long term information storage capability ; Strong interpretability | More complex model design relatively |
> > >
> > > **3- For what specific scenarios or use cases is each type of memory best suited?**
> > >
> > > Implicit memory is more suitable for on-cloud applications (e.g., large language models hosted on large GPU clusters) where memory constraints are not a concern, so we could directly cache all past tokens to preserve information.
> > > Explicit memory is better suited for on-edge applications with limited computation resources, enabling more effective storage of past information through explicit representations.
> > > In scenarios that require balancing storage compactness and long-term information preservation (e.g., streaming 3D reconstruction), explicit memory usually has an advantage due to its more compact and comprehensive information storage mechanism.
> > >
> > > We hope the above response can help address your concerns. We are happy to answer any additional questions you may have.

---

> > > > ### Comment · Reviewer_JUWw · 2025-08-06
> > > >
> > > > Thank you for your work. I have a few remaining questions and points of clarification that I would like to address.
> > > >
> > > >
> > > >
> > > > ### 1. Handling of Dynamic Scenes and the Memory Fusion Mechanism
> > > >
> > > >
> > > >
> > > > The proposed memory fusion mechanism appears to be a core component of your model. However, I have concerns about its robustness in **dynamic scenes**, an issue also highlighted by Reviewer nKKa regarding "point drifting."
> > > >
> > > > For instance, consider a scenario where a person is visible in one area of a room in an early frame, but is no longer present when the camera returns to that same viewpoint in a later frame.
> > > >
> > > > - How does your current fusion mechanism handle such temporal inconsistencies? Does it risk retaining outdated information, potentially leading to a corrupted or inaccurate final scene representation?
> > > > - Given this challenge, do you believe a different fusion strategy might be necessary for dynamic environments? For example, the implicit architecture of a model like `StreamVGGT` seems inherently better suited for these scenarios. Could you comment on this comparison?
> > > >
> > > > ------
> > > >
> > > >
> > > >
> > > > ### 2. Detailed Comparison with State-of-the-Art Methods
> > > >
> > > >
> > > >
> > > > To better understand the contributions and positioning of your work, I would appreciate a more detailed comparison with related models.
> > > >
> > > > - **Comparison with `Point3D` and `StreamVGGT`**: Could you provide a comprehensive analysis comparing your model to `Point3D` and `StreamVGGT` across the following aspects?
> > > >   - **Model Architecture**: Key structural differences and design insights.
> > > >   - **Target Scenarios**: The types of scenes or applications where each model excels.
> > > >   - **Training Data**: Differences in data requirements or pre-training strategies.
> > > >   - **Advantages & Disadvantages**: A frank assessment of the relative strengths and weaknesses.
> > > > - **Benchmark Comparison with `Mast3r-SLAM`**: To empirically situate your method's performance, could you provide a direct comparison against `Mast3r-SLAM` using its own established benchmarks? The comparison should ideally cover:
> > > >   - **Mapping Quality**
> > > >   - **Tracking Accuracy**
> > > >   - **Computational Efficiency**
> > > >
> > > > ------
> > > >
> > > > I believe that clear and objective answers to these questions during the discussion period would significantly clarify the merits and limitations of your work. **I will raise my score to 5 if my questions are fully and objectively answered.**

---

> > > > > ### Author Response · Authors · 2025-08-07
> > > > >
> > > > > We thank the reviewer for the timely reply and constructive suggestions, which are greatly helpful in improving our paper.
> > > > > We appreciate the opportunity to further clarify your concerns.
> > > > >
> > > > > **1- Handling of Dynamic Scenes and the Memory Fusion Mechanism**
> > > > >
> > > > > Thanks for this valuable question.
> > > > > When dealing with dynamic scenes, how to update the memory is a crucial problem.
> > > > > We use a memory encoding module to obtain new pointers which represent the scene features of their corresponding local regions at the current time.
> > > > > Then, we perform a nearest neighbor search in the memory for each new pointer.
> > > > > If the distance is sufficiently small, we assume that the new pointer and its nearest neighbor (in the memory) refer to the same spatial region.
> > > > > In that case, we fuse their positions and features, ensuring that the new memory pointer encode information from both past frames and the latest observation.
> > > > > In other words, after each update, every pointer in the memory corresponds to the most recent scene information around its associated spatial region.
> > > > > This is exactly why our method can also handle dynamic scenes.
> > > > >
> > > > > StreamVGGT caches the historical tokens frame by frame and conducts temporal causal attention to enable the reconstruction from streaming inputs.
> > > > > However, this direct storage of implicit features from past frames ignores their spatial relationships in the current scene, leading to increasingly redundant information and higher computational overhead as the sequence grows.
> > > > > Moreover, storing all past information makes the reconstruction of the latest frame more susceptible to the influence of earlier frames that are temporally distant (the temporal inconsistencies you have mentioned).
> > > > >
> > > > > In rare but challenging cases where the spatial content at a location changes drastically within a short time span (a challenge for almost all dynamic streaming-based methods), adjusting the weighting between current and past frame information dynamically in the memory fusion module would be better suited to handle such scenarios.
> > > > > We think it is an interesting and important direction to improve our method in the future.
> > > > >
> > > > > **2- Detailed Comparison with State-of-the-Art Methods**
> > > > >
> > > > > **Comparison with Point3D and StreamVGGT:**
> > > > >
> > > > > |Method|Model Architecture|Target Scenarios|Training Data|Advantages|Disadvantages|
> > > > > |:-|:-|:-|:-|:-|:-|
> > > > > |Point3D|Anchor-free architecture with a point head to localize the action and time-wise attention to intergrate context information|Spatio-temporal action recognition|3 action-recognition datasets to train|Stable and accurate action recognition in videos|Can not obtain dense geometry results|
> > > > > |StreamVGGT|Cached token memory and temporal causal attention|Streaming dense reconstruction without considering memory consumption|A knowledge distillation strategy to finetune on 13 datasets|Fast inference and dense geometry outputs|Increasing memory consumption as the sequence grows|
> > > > > |Ours|Explicit pointer memory and pointer-image interaction|Streaming dense reconstruction balancing memory consumption and long-term accuracy|Train from scratch on 14 datasets|Accurate and dense geometry reconstrcution on long videos with low memory usage|Relatively low inference speed|
> > > > >
> > > > > **Benchmark Comparison with MASt3R-SLAM:**
> > > > > Thanks for your valuable advice.
> > > > > We compare our method with MASt3R-SLAM on its own established benchmarks and the results are below.
> > > > >
> > > > > We conducted the dense geometry evaluation (**Mapping Quality**) on 7-Scenes seq-01 and report the average results across different scenes in the following table.
> > > > >
> > > > > |Method|Accuracy ↓|Completion ↓|Chamfer ↓|
> > > > > |:-:|:-:|:-:|:-:|
> > > > > |MASt3R-SLAM|0.068|0.045|0.056|
> > > > > |**Ours**|**0.061**|**0.022**|**0.042**|
> > > > >
> > > > > We also report the RMSE of the absolute trajectory error (ATE ↓, **Tracking Accuracy**) in meters on 7-Scenes and TUM RGB-D.
> > > > >
> > > > > |Method|Chess|Fire|Heads|Office|Pumpkin|Redkitchen|Stairs|Avg|
> > > > > |:-:|:-:|:-:|:-:|:-:|:-:|:-:|:-:|:-:|
> > > > > |MASt3R-SLAM|0.063|0.046|0.029|0.103|0.114|0.074|0.032|0.066|
> > > > > |Ours|0.076|0.055|0.036|0.125|0.094|0.068|0.065|0.084|
> > > > >
> > > > > |Method|360|Desk|Desk2|Floor|Plant|Room|Rpy|Teddy|Xyz|Avg|
> > > > > |:-:|:-:|:-:|:-:|:-:|:-:|:-:|:-:|:-:|:-:|:-:|
> > > > > |MASt3R-SLAM|0.070|0.035|0.055|0.056|0.035|0.118|0.041|0.114|0.020|0.060|
> > > > > |Ours|0.114|0.071|0.079|0.085|0.047|0.105|0.058|0.115|0.035|0.084|
> > > > >
> > > > > Finally, we compare the peak memory usage and per-frame rumtime (**Computational Efficiency**) between MASt3R-SLAM and our method on long sequences sampled from 7-Scenes in the table below.
> > > > >
> > > > > |Method|Peak Memory Usage|Per-frame Runtime|
> > > > > |:-:|:-:|:-:|
> > > > > |MASt3R-SLAM|7.18 GB|0.11 s|
> > > > > |Ours|5.46 GB|0.20s|
> > > > >
> > > > > We can observe that our method achieves better dense geometry reconstruction results with lower memory consumption.
> > > > > However, there still has room for further improvement in tracking accuracy (as discussed in our Limitations) and runtime efficiency (as we have discussed in the earlier comments).
> > > > >
> > > > > We hope the above response can help address your concerns. We are happy to answer any additional questions you may have.

---

> > > > > > ### Comment · Reviewer_JUWw · 2025-08-07
> > > > > >
> > > > > > Thanks to the authors for further clarification. My concerns are addressed. I will raise score to 5.

---

> ### Comment · Reviewer_JUWw · 2025-08-08
> **On the Fundamental Merit of the Explicit Memory Paradigm for Long-Term Consistency**
>
> Dear Authors,
>
> Thank you for your contribution to streaming 3D reconstruction with the Point3R framework.
>
> The paper contrasts Point3R with other memory-based approaches, highlighting the benefits of its explicit 3D memory. However, it is important to consider that state-of-the-art SLAM systems (e.g., a hypothetical MASt3R-SLAM, which leverages a powerful front-end) also maintain a form of explicit 3D memory. This memory typically exists as a keyframe-based pose graph with associated 3D map points, and more crucially, it explicitly ensures long-term global consistency through backend optimization (e.g., global Bundle Adjustment after a loop closure is detected).
>
> Point3R, in contrast, employs a learned, continuous fusion mechanism to integrate new information into its unified memory on a frame-by-frame basis. While this may offer greater flexibility for handling local scene changes, minor errors can inevitably accumulate over extended trajectories.
>
> This leads to my question:
>
> 1. **Compared to a traditional SLAM system like MASt3R-SLAM, which enforces global consistency by explicitly correcting accumulated errors through non-local, global optimization, what is the fundamental merit of Point3R's learned, frame-by-frame fusion approach in ensuring long-term consistency for large-scale scene reconstruction? Could you further elaborate on its core value compared to MASt3R-SLAM by considering various aspects?**
>
> 2. Besides, when a scene is large enough and a trajectory is long enough for loop closures to occur, how does Point3R's architecture address the accumulated drift that a classic SLAM system would resolve through global optimization? In what specific scenarios or tasks do you believe Point3R's "continuous fusion" paradigm would demonstrate a decisive advantage over the classic "track-map-optimize" paradigm?
>
> 3. **Please provide a comparative analysis of MASt3R-SLAM and Point3R, assessing their mapping quality, tracking accuracy, and computational efficiency on short, medium, and long sequences. Your report should also include the methodology, detailing how these sequence lengths are defined and how they were generated from the source data.**

---

> > ### Author Response · Authors · 2025-08-09
> > **Comment-Part 1**
> >
> > Thanks for the questions. We are happy to provide more discussions.
> >
> > Firstly, we think traditional SLAM systems are very good and more mature solutions to obtain relatively accurate camera pose from videos.
> > However, we would like to point out that the core objectives of SLAM and the recent feed-forward 3D reconstruction methods (e.g., DUSt3R, CUT3R, and ours) are different.
> > The main objective of SLAM lies in accurately registering the camera poses from RGB-D inputs (where the depth could be obtained from prediction models). Differently, feed-forward 3D reconstruction methods aim to develop a unified model to predict 3D reconstructed points for each frame in a shared coordinate system (and some recent methods simultaneously predict the camera pose with an additional head).
> >
> > Note that **the two types of models are compatible and essentially beneficial to each other**.
> > Visual SLAM systems could use a feed-forward 3D reconstruction model to produce reconstructed points.
> > Feed-forward 3D reconstruction methods can further employ SLAM techniques such as bundle adjustment to obtain more accurate poses.
> >
> > **1-Comparison of MASt3R-SLAM and Point3R**
> > A traditional SLAM system like MASt3R-SLAM relies on global optimization to ensure long-term consistency.
> > Specifically, MASt3R-SLAM builds on the output of MASt3R (a data-driven feed-forward paradigm designed for pairwise image matching) and then uses global optimization to obtain more accurate pose estimates.
> > However, it is worth noting that such accuracy is still fundamentally based on the local outputs of MASt3R’s feed-forward approach.
> > In fact, MASt3R-SLAM precisely demonstrates that the feed-forward reconstruction paradigm, when integrated with the global optimization concepts from the SLAM system, can achieve improved performance in accurate localization and mapping.
> >
> > Our core objective is not to present a fully developed system and outperform SLAM methods.
> > The core contribution of our work lies in how to improve the feed-forward paradigm to better handle the streaming online setting to mitigate long-term error accumulation.
> > We believe this is meaningful, as the recent feed-forward 3D reconstruction methods are generalizable methods which can benefit from more training data.
> > For this emerging field, we think the potential to scale up with more data is important to build a more generalizable "large geometry model" similar to large language models.
> >
> > **2-Mitigation of long-term error accumulation**
> > We mitigate the error accumulation by extending the memory to try to preserve more direct information. However, directly preserving all the past tokens would cause a very heavy computation burden over time, and there is then a trade-off between information preserving and efficiency. The objective of using explicit 3D points is then to improve this trade-off by merging tokens that are close in the 3D space. This can keep the total number of memory tokens within a reasonable range while covering the observed area. We update the pointer memory to incorporate the newest information. Note that some memories are not updated (unobserved by the current frame) and can be directly accessed to mitigate the error accumulation.
> >
> > We clarify again that visual SLAM and feed-forward 3D reconstruction are compatible with each other.
> > We further analyze the advantages and disadvantages of them in the following table.
> >
> > |Method|Advantages|Disadvantages|
> > |:-|:-|:-|
> > |Visual SLAM|Higher pose estimation accuracy|Complex pipeline; Dependency on 3D reconstruction models|
> > |Ours|End-to-end paradigm to simultaneously produce both dense reconstructed points and camera poses with per-frame streaming RGB input; No dependency on any other modules|Still has room for improvement in estimation accuracy (without post process)|
> >
> > Please check Comment-Part 2 for more responses. Thank you.

---

> > ### Author Response · Authors · 2025-08-09
> > **Comment-Part 2**
> >
> > **3-Comparative analysis of MASt3R-SLAM and Point3R**
> >
> > Following this suggestion, we sampled short, medium, and long sequences from 7-scenes seq-01.
> > Specifically, the short sequences were sampled at an interval of 50, resulting in 20 frames per sampled sequence.
> > The medium sequences were sampled at an interval of 10, resulting in 100 frames per sampled sequence.
> > The long sequences were sampled at an interval of 2, resulting in 500 frames per sampled sequence.
> >
> > We conducted the dense geometry evaluation (**Mapping Quality**) and report the average results across different scenes in the following table.
> >
> > |Method|Length|Accuracy ↓|Completion ↓|Chamfer ↓|
> > |:-:|:-:|:-:|:-:|:-:|
> > |MASt3R-SLAM|Short|0.031|0.027|0.029|
> > |Ours|Short|0.034|0.018|0.026|
> > |MASt3R-SLAM|Medium|0.038|0.029|0.034|
> > |Ours|Medium|0.037|0.019|0.028|
> > |MASt3R-SLAM|Long|0.068|0.045|0.056|
> > |Ours|Long|0.061|0.022|0.042|
> >
> > We also report the RMSE of the absolute trajectory error (ATE ↓, **Tracking Accuracy**) in meters.
> >
> > |Method|Length|Chess|Fire|Heads|Office|Pumpkin|Redkitchen|Stairs|Avg|
> > |:-:|:-:|:-:|:-:|:-:|:-:|:-:|:-:|:-:|:-:|
> > |MASt3R-SLAM|Short|0.059|0.046|0.014|0.106|0.166|0.080|0.230|0.100|
> > |Ours|Short|0.054|0.067|0.029|0.090|0.163|0.046|0.210|0.094|
> > |MASt3R-SLAM|Medium|0.062|0.036|0.034|0.098|0.114|0.076|0.062|0.069|
> > |Ours|Medium|0.056|0.047|0.040|0.124|0.090|0.054|0.067|0.068|
> > |MASt3R-SLAM|Long|0.063|0.046|0.029|0.103|0.114|0.074|0.032|0.066|
> > |Ours|Long|0.076|0.055|0.036|0.125|0.094|0.068|0.065|0.084|
> >
> > And finally, we compare the peak memory usage and per-frame runtime (**Computational Efficiency**) between MASt3R-SLAM and our method.
> >
> > |Method|Length|Peak Memory Usage|Per-frame Runtime|
> > |:-:|:-:|:-:|:-:|
> > |MASt3R-SLAM|Short|7.18 GB|0.17 s|
> > |Ours|Short| 4.02 GB| 0.20 s|
> > |MASt3R-SLAM|Medium|7.18 GB|0.15 s|
> > |Ours|Medium|4.99 GB|0.20s|
> > |MASt3R-SLAM|Long|7.18 GB|0.11 s|
> > |Ours|Long|5.46 GB|0.20s|
> >
> > We hope our discussions are helpful. We are happy to answer any additional questions you may have.

---

> ### Comment · Reviewer_JUWw · 2025-08-09
>
> Dear Authors,
>
> Thanks for your work.
>
> In the future, do you think an explicit 3D memory bank is necessary? It seems that scalable data, combined with an implicit memory bank that encodes spatial information, could solve the problem of long-term 3D information storage.
>
> Why would it be necessary? Please provide your reasons.
> Why would it be unnecessary? Please provide your reasons.
> Alternatively, from what perspectives should we consider its necessity or lack thereof?
>
> For example, Google's recent model, Genie, appears to achieve excellent geometry consistency through scalability without explicit 3D constraints. Does this imply that for a task like online 3D/4D reconstruction, an explicit 3D memory bank is also unnecessary?
>
> **If we were to argue for the necessity of an explicit 3D memory bank, could it (e.g., Point3r) be understood as a form of trade-off compression between an implicit 3D memory bank (e.g., CUT3r) and a fully explicit 3D representation (e.g., Mast3r-SLAM)?**

---

> > ### Author Response · Authors · 2025-08-09
> >
> > Thanks for the questions. We are happy to provide more discussions.
> >
> > We also notice Google's newest model Genie 3 (released four days ago) has demonstrated excellent 3D spatial consistency.
> > They did not share the model architecture, so it is still unknown whether they use an explicit 3D representation or not.
> >
> > That being said, we can still infer its architecture based on its behavior. We notice that Genie 3 can achieve **real-time** and **long-term** generation. This implies a possible autoregressive diffusion architecture with a **compressed memory mechanism** to store past information. We also notice that it can achieve good long-term spatial and appearance consistency. This implies possibly **long-term memory preserving of both spatial and semantic information**.
> >
> > We think the key design of Genie 3 is an efficient memory mechanism with a good trade-off between information preserving and memory compactness. This is exactly the key motivation for our method. We thus speculate that Genie 3 employs a similar explicit 3D memory bank as ours. In this way, the memory can preserve compact and comprehensive information of past observations/generations of both spatial positions and semantic features. Possibly, this is achieved by fusing redundant tokens by merging tokens (both positions and semantic features) that are close in the 3D space.
> >
> > Therefore, we think it is necessary for an explicit 3D memory bank.
> > We agree with the reviewer's understanding that our design can be understood as a form of trade-off compression between an implicit 3D memory bank and a fully explicit 3D representation.
> > We think it is the key to achieve **real-time** streaming prediction/generation with a good **long-term consistency**.
> >
> > We hope our discussions are helpful. We are happy to answer any additional questions you may have.

---

> > > ### Comment · Reviewer_JUWw · 2025-08-09
> > >
> > > Dear Authors,
> > >
> > > Thank you for your response.
> > >
> > > We were wondering if you have any further suggestions for mitigating the coordinate system challenges (the first frame is the anchor frame) in **online 3D/4D reconstruction**. We've observed this to be a common issue, affecting models like Dust3r, Spann3r, Cut3r, and VGGT.
> > >
> > > Any additional insights you could offer would be greatly appreciated.
> > >
> > > Best regards,

---

### Official Review · Reviewer_nKKa · 2025-07-01

**Clarity:** 3
**Significance:** 3
**Originality:** 2
**Rating:** 4
**Confidence:** 3

**Summary:**

This work extends DUSt3R with a memory mechanism. The memory takes the form of a set of featured 3D points in a global coordinate system. The memory system allows the pipeline to replace the expensive cross-attention between frames, commonly used with interactions between image features and memory, making processing sequential input much more efficient. Experiment results show that the performance is comparable to baselines.

**Questions:**

See weakness.

I am concerned about memory points drifting. The memory fusion seems to be rule-based, hard association. I'd assume that given certain input sequence, the memory points may drift after each update. In the worst case, some memory points may be carried from the very first frame to the very last frame. I am wondering why a more elaborate approach is not needed in this case.

**Ethical Concerns:**

["NO or VERY MINOR ethics concerns only"]

**Final Justification:**

While the evaluation presented in the main paper did not demonstrate significant advantages, I have raised my score after reviewing the results on long sequences provided during the rebuttal. These additional results show that incorporating explicit memory can indeed improve reconstruction quality. That said, I believe a more comprehensive discussion of these findings should be included in the main paper.

**Limitations:**

The limitations are discussed albeit briefly.

**Quality:**

3

**Strengths And Weaknesses:**

Strengths: The idea is solid and novel. Removing the interaction between frames can greatly improve the scalability. The experimental results show only a slight performance drop on certain datasets and are still in a comparable range.


Weaknesses: 1) I find it hard to determine the novelty of 3D hierarchical positional embedding. According to my understanding, the hierarchical positional embedding refers to computing RoPE using different frequency bases than taking the average. While the author emphasized this as a major contribution in the abstract and devoted a whole section to it, I find it more like an engineering detail. While in table 5, the authors provide the results with and w/o 3DHPE, it is not clear to me what is w/o 3DHPE. 2) It seems that the main advantage is efficiency. However, the advantage is not emphasized in experiment section. The majority of the experiment section is devoted to showing that the performance is only comparable to previous works. Thus, I think the main contribution is not well supported.

---

> ### Author Rebuttal · Authors · 2025-07-31
>
> Thanks for your thoughtful comments, questions and insightful suggestions.
> We provide detailed responses below.
>
> **1 - Novelty of 3D hierarchical positional embedding**
>
> The transformer architecture exhibits a lack of order among tokens, necessitating positional embedding to incorporate relational priors.
> Therefore, the design of a 3D positional embedding lies in the core of our method for explicit spatial modeling.
> However, the extension of RoPE is not straightforward and poses some challenges.
> - Unlike the discrete positions used by RoPE, 3D spatial priors usually require more accurate continuous positions.
> - We use a variety of training datasets that exhibit large scale variations in the input coordinates.
>
> To address this challenge, we design our 3D hierarchical positional embedding with the following intuitions:
> - We directly encode continuous 3D coordinates.
> - We employ hierarchical frequency bases to make our positional embedding more flexible and adaptive to different scales.
> By rotating tokens under multiple frequency bases and averaging the results, we ensure that at least one frequency-aligned subspace reliably encodes the relative positional relationship, regardless of the coordinate scale.
>
> We think the novelty of a positional embedding method lies in the underlying intuitions of the design instead of the implementation (which usually turns out to be simple yet effective).
>
> We apologize for the confusion in Table 5.
> "w/o 3DHPE" means we remove this 3D hierarchical position embedding in our pointer-image interaction during the training and evaluation.
> To further demonstrate the effectiveness of our 3D hierarchical positional embedding, we conducted more comprehensive ablation studies.
> Specifically, we implemented the 3D position embedding using a single frequency base (i.e., removing the hierarchical design), retrained the model, and evaluated its 3D reconstruction performance ("w/o hierarchical freq" in the table below).
> As the results show, applying no embedding in the pointer-image interaction, using a single-frequency embedding, and adopting our 3D hierarchical embedding lead to progressively improved reconstruction performance.
>
> | Method | Dataset | Acc ↓ | Comp ↓ | NC ↑ | Dataset | Acc ↓ | Comp ↓ | NC ↑ |
> |:------:|:-----:|:-----:|:-----:|:-----:|:-----:|:-------:|:--------:|:--------:|
> | Ours (w/o 3DHPE) | 7-scenes | 0.180 | 0.180 | 0.683 | NRGBD | 0.145 | 0.123   | 0.770    |
> | Ours (w/o hierarchical freq) | 7-scenes | 0.130 | 0.143 | 0.722 | NRGBD | 0.085 | 0.080   | 0.819    |
> | **Ours (w 3DHPE)** | 7-scenes | **0.124** | **0.139** | **0.725** | NRGBD | **0.079** | **0.073**   | **0.824**    |
>
> **2 - Main advantage & discussion on efficiency**
>
> Sorry for the confusion.
> To be more accurate, the main motivation is the efficiency to preserve more information when handling long sequences, resulting from the explicit modeling of memory tokens.
> CUT3R employs a fixed-size memory and replaces the previous one with the updated one (implicitly obtained by the most recent observation), which may suffer from the loss of early-frame information when processing long sequences.
> A straightforward solution is to keep all past memories instead of replacing them.
> However, as the number of frames increases, such storage becomes undoubtedly redundant and inefficient.
> This motivates us to explore how to effectively fuse these redundant tokens, and we achieve this by merging tokens that are close in the 3D space.
>
> The results below show the efficiency improvement when using our explicit memory fusion module by comparing the per-frame runtime and the peak memory usage.
>
> | Method | #Frames | Runtime (per frame) | Memory Usage (peak) | #Frames | Runtime (per frame) | Memory Usage (peak) | #Frames | Runtime (per frame) | Memory Usage (peak) |
> |:------:|:-------:|:-----:|:-----:|:-------:|:-----:|:-----:|:-------:|:-----:|:-----:|
> | Ours (w/o Mem-Fusion) | 10 | 0.25 s | 3.81 GB | 20 | 0.38 s | 4.52 GB | 30 | 0.66 s | 5.23 GB |
> | Ours | 10 | 0.20 s | 3.69 GB | 20 | 0.20 s | 4.02 GB | 30 | 0.21 s | 4.47 GB |
>
> Furthermore, to better valid the advantage of our explicit memory to deal with the loss of early-frame information during long-sequence reconstruction, we sample a sequence with a frame interval of 10 (each sequence contains 50-100 frames) for each scene in 7-scenes, and sample a sequence with a frame interval of 20 (each sequence contains 40-80 frames) for each scene in NRGBD to compare the long sequence reconstruction performance with CUT3R.
>
> | Method | Dataset | Acc (mean) ↓ | Acc (med) ↓ | Comp (mean) ↓ | Comp (med) ↓ | NC (mean) ↑ | NC (med) ↑ |
> |:------:|:-------:|:-----:|:-----:|:-----:|:-----:|:-------:|:--------:|
> | CUT3R  | 7-scenes | 0.038 | 0.019 | 0.029 | 0.008 | 0.623   | 0.693    |
> | **Ours**   | 7-scenes | **0.035** | **0.017** | **0.024** | **0.007** | **0.627**   | **0.701**    |
> | CUT3R  | NRGBD   | 0.093 | 0.043 | 0.037 | 0.010 | 0.744   | 0.906    |
> | **Ours**   | NRGBD   | **0.073** | **0.034** | **0.026** | **0.008** | **0.758**   | **0.917**    |
>
>
> Also,  due to the use of explicit spatial priors, our model shows higher **training efficiency**, i.e., our model achieves comparable performance with significantly lower training cost.
> Our model was trained for 7 days on 8 GPUs (while CUT3R required around a month) and used fewer than half of the training datasets used in CUT3R.
> We think this **training efficiency** is also valuable for promoting research in this field.
>
> **3 - Memory points drifting**
>
> Thanks for this valuable question.
> We agree that the memory points may drift after each update since we update their positions.
> However, note that we fuse both the 3D coordinates and also the **features**.
> The updated memory points then represent the information of the updated position.
> The memory points drifting is therefore an intended phenomenon for dynamic management of the memory.
>
> We hope the above response can help address your concerns. We are happy to answer any additional questions you may have.

---

> > ### Comment · Reviewer_nKKa · 2025-08-04
> > **Official Comment**
> >
> > I thank the authors for provideing additional results.
> > Below are some of addtional questions.
> >
> > 1. According to my understanding, the main contribution of this work is efficiency.
> > I'd expect to see a full subsection dedicated to comparsion with other baselines like CUT3R in terms of runtime, memory, the amount of data used, etc. Currently, from both the paper and rebuttal, I didn't see a comperhensive discussion.
> >
> > 2.Won't point drafting leave a holes in the scene and lead to loss of information especially when the number of pointers is low?
> >
> > 3. For training datasets with large scale variations in the input coordinates, did you consider normalizing the coordinates? Or is there a specific reason why normalization would be difficult or inappropriate in this context? Do other models also suffer from the same issue?

---

> > > ### Author Response · Authors · 2025-08-05
> > >
> > > We thank the reviewer for the timely reply and constructive suggestions, which are greatly helpful in improving our paper. We appreciate the opportunity to further clarify your concerns.
> > >
> > > **1 - Main contribution**
> > >
> > > We apologize for the confusion. We would like to clarify that the main contribution of our work is **not** the efficiency as the reviewer may understand. The efficiency discussed in the paper mainly refers to the efficiency of memory-based methods (e.g., CUT3R) over pair-wise methods (e.g., DUSt3R), since they can achieve online frame-by-frame reconstruction. However, one disadvantage of memory-based methods is the lack of comprehensive representation of past observations from the memory. This results in a trade-off between information preserving and memory compactness. That is, we can preserve more information by increasing the memory size, but this will lead to more computation in the interaction between the current input and the memory. **Our main contribution is then how to improve this trade-off.** Our solution is to employ explicit spatial memory tokens, so we can effectively fuse the redundant tokens that are close in the 3D space. This promotes the memory tokens to tend to uniformly distributed in the observed 3D space, and thus achieves a good trade-off between comprehensiveness and compactness.
> > >
> > > The efficiency of our method is the lower runtime and memory consumption over the counterpart of preserving all past tokens as the memory (without our spatial-aware memory fusion), as verified in the second table in our previous rebuttal. Note the runtime tends to stabilize after a certain number of frames (approximately 0.2s) when the memory tokens are enough to describe the observed 3D space. We will improve relevant sentences in the paper to avoid ambiguity.
> > >
> > > We mainly compared the performance with the other baselines (e.g., CUT3R) to show the ability of our method to preserve more information. This is also further verified in the long-sequence settings, as shown in the third table in our previous rebuttal.
> > >
> > > For the data and training efficiency, we further supplement the following table.
> > > We provide a comparison between our method and CUT3R on the number of training datasets and the number of training sequences sampled per epoch at different training stages.
> > >
> > > |Method|Stage|Training datasets|Sample size|Stage|Training datasets|Sample size|Stage|Training datasets|Sample size|Stage|Training datasets|Sample size|Total training time|
> > > |:------:|:-------:|:-----:|:-----:|:-------:|:-----:|:-----:|:-------:|:-----:|:-----:|:-------:|:-----:|:-----:|:-----:|
> > > |CUT3R|Stage 1|14|806000|Stage 2|31|801880|Stage 3|30|729248 | Stage 4 | 25 | 398684 | 30 days |
> > > |Ours|Stage 1|10|100000|Stage 2|14|118000|Stage 3|14|33800 | - | - | - | 7 days |
> > >
> > > We think these experiments are sufficient to verify our contribution of improving the trade-off between information preserving and memory compactness for memory-based methods. Still, feel free to request more information you think is needed, and we are more than happy to discuss more and provide more results.
> > >
> > > **2 - Point drifting**
> > >
> > > During each update, we fuse the new pointer with its nearest neighbor in the memory only when their distance is below a certain threshold $\delta$.
> > > This means that each pointer in the memory undergoes only a minor adjustment ($\delta$) in its position and is unlikely to drift too far.
> > > New points that are beyond this distance threshold $\delta$ would result in new pointers to expand the memory.
> > > Our memory encoding and fusion module is designed so that all pointers in memory tend to evenly distributed across the explored regions, avoiding over-concentration or holes.
> > > To further verify this, we conducted a statistical analysis across all testing scenes in the NRGBD dataset.
> > > After processing each sequence, we computed the distance from each ground-truth point to its nearest neighbor pointer in the memory, and analyzed the maximum of these distances (i.e., the largest size of a "hole").
> > > We find that this value is below $1.5 \delta$ in all the testing sequences, showing that all ground-truth points are effectively covered by their nearby pointers.
> > >
> > > **3 - Scale normalization**
> > >
> > > Thanks for your question.
> > > We did not normalize the coordinates since we want the model to reconstruct 3D points with real-world scales.
> > > Some methods (e.g., DUSt3R) normalize both the predicted and ground-truth pointmaps accordingly to encourage the model to learn scale-invariant reconstruction.
> > > In contrast, we leverage the metric ground-truth annotations (when available) to guide the model toward producing outputs that align with real-world scales, making it more applicable to use-cases that require metric-scale predictions.
> > > Several other methods (e.g., MASt3R and CUT3R) also do not normalize the coordinates when metric annotations are available.
> > >
> > > We hope the above response can help address your concerns. We are happy to answer any additional questions you may have.

---

### Official Review · Reviewer_8Vja · 2025-07-02

**Clarity:** 3
**Significance:** 2
**Originality:** 3
**Rating:** 4
**Confidence:** 4

**Summary:**

This paper presents Point3R, a novel streaming 3D reconstruction method that maintains explicit point memory. Point3R decodes camera parameters and point maps by interacting current image features with memory. Additionally, after decoding, the memory is updated by fusing the previous nearest neighbor pointers. The authors validate their proposed method on depth estimation, camera estimation, and 3D reconstruction tasks.

**Questions:**

See the weaknesses, especially the first question: why is explicit pointer memory superior?

**Ethical Concerns:**

["NO or VERY MINOR ethics concerns only"]

**Final Justification:**

The authors provide additional comparison with SOTA methods such as VGGT and most of my concerns are addressed.

**Limitations:**

Yes

**Quality:**

2

**Strengths And Weaknesses:**

Strengths:
1. The analysis of various 3D reconstruction methods is insightful. Point3R and other streaming reconstruction techniques are particularly suited for embodied systems.

2. The introduction of the 3D Rope concept into the pointer memory interaction process is well-designed. The approach for assigning 3D positions $p(u,v)$ for image patches is reasonable.

3. The presentation is clear and easy to follow.

Weaknesses:
1. The authors assert that explicit position memory is superior, which is central to the Point3R concept. However, a convincing analysis is lacking. Why is storing features in an explicit position advantageous? From my perspective, the manually designed explicit memory fusion and encoding strategy appears coarse and prone to information loss. For instance, the explicit memory updating relies on nearest neighbor searching and average pooling, whereas the implicit memory updating method in CUT3R is fully data-driven and utilizes flexible attention. Moreover, experimental results do not support the core claim, as CUT3R outperforms Point3R in several tasks.

2. Point3R is not compared with VGGT, a state-of-the-art “All-to-All” interaction method mentioned in the introduction. A comparison with VGGT is expected and would strengthen the paper.

3. The authors do not provide convincing ablation studies. Key designs, such as the 3D Rope and explicit pointers, lack validation.

CUT3R [Wang Q, Zhang Y, Holynski A, et al. Continuous 3D Perception Model with Persistent State[J]. arXiv preprint arXiv:2501.12387, 2025.]

VGGT [Wang J, Chen M, Karaev N, et al. Vggt: Visual geometry grounded transformer[C]//Proceedings of the Computer Vision and Pattern Recognition Conference. 2025: 5294-5306.]

---

> ### Author Rebuttal · Authors · 2025-07-31
>
> Thanks for your thoughtful comments, questions and insightful suggestions.
> We provide detailed responses below.
>
> **1 - More analysis about the explicit pointer memory**
>
> Thanks for the suggestion.
> We store features in an explicit position to enable spatial-aware memory management to improve the trade-off between information preserving and efficiency.
> CUT3R employs a fixed-size memory and replaces the previous one with the updated one (implicitly obtained by the most recent observation), which may suffer from the loss of early-frame information when processing long sequences.
> A straightforward solution is to keep all past memories instead of replacing them.
> However, as the number of frames increases, such storage becomes undoubtedly redundant and inefficient.
> This motivates us to explore how to effectively fuse these redundant tokens, and we achieve this by merging tokens that are close in the 3D space.
> This requires the storage of each token in an explicit position.
> We then design a simple yet effective token merging strategy (nearest neighbor searching and average pooling), which can keep the total number of memory tokens within a reasonable range.
> The obtaining of the memory token in our model is also data-driven and flexible, the explicit position modeling and memory updating mechanism are mainly for managing the memory size while preserving the most important information.
>
> To further support this, we compare our performance with CUT3R under a long-sequence reconstruction setting as shown in the table below.
>
> | Method | Dataset | Acc (mean) ↓ | Acc (med) ↓ | Comp (mean) ↓ | Comp (med) ↓ | NC (mean) ↑ | NC (med) ↑ |
> |:------:|:-------:|:-----:|:-----:|:-----:|:-----:|:-------:|:--------:|
> | CUT3R  | 7-scenes | 0.038 | 0.019 | 0.029 | 0.008 | 0.623   | 0.693    |
> | **Ours**   | 7-scenes | **0.035** | **0.017** | **0.024** | **0.007** | **0.627**   | **0.701**    |
> | CUT3R  | NRGBD   | 0.093 | 0.043 | 0.037 | 0.010 | 0.744   | 0.906    |
> | **Ours**   | NRGBD   | **0.073** | **0.034** | **0.026** | **0.008** | **0.758**   | **0.917**    |
>
> Specifically, for each scene in 7-scenes, we sample a sequence with a frame interval of 10 (each sequence contains 50-100 frames).
> For each scene in NRGBD, we sample a sequence with a frame interval of 20 (each sequence contains 40-80 frames).
> We see that our model outperforms CUT3R in this setting.
>
> Also note that our model achieves comparable performance with significantly lower training cost due to the explicit spatial prior.
> Our model was trained for 7 days on 8 GPUs (while CUT3R required around a month) and used fewer than half of the training datasets used in CUT3R.
> This demonstrates the advantage of our proposed explicit pointer memory to enable the model to learn and generalize more effectively.
>
>
> **2 - Comparison with VGGT**
>
> Thanks for the suggestion.
> As noted by the official NeurIPS guidelines ("papers that appeared online after March 1st, 2025 will generally be considered 'contemporaneous'"), we considered VGGT as a contemporaneous work (public on arXiv on March 14th) and therefore only discussed but did not compare intensely in the paper.
> Also, our model targets the streaming frame-by-frame online reconstruction setting while VGGT targets one-shot global reconstruction with all inputs, which usually achieves better performance at the cost of efficiency.
>
> In this rebuttal, we therefore compare with StreamVGGT [1] (a streaming version of VGGT) to analyze the advantage of our method in the streaming setting and only provide the results of VGGT as a reference.
>
> | Method | Setting | Dataset | Interval | Acc (mean) ↓ | Acc (med) ↓ | Comp (mean) ↓ | Comp (med) ↓ | NC (mean) ↑ | NC (med) ↑ |
> |:------:|:------:|:-------:|:-----:|:-----:|:-----:|:-----:|:-----:|:-------:|:--------:|
> | VGGT  | Global | 7-scenes | 10 | 0.017 | 0.006 | 0.028 | 0.010 | 0.651   | 0.735    |
> | StreamVGGT | Streaming | 7-scenes | 10 | 0.043 | 0.022 | 0.028 | 0.009 | 0.647   | 0.730    |
> | **Ours**  | Streaming | 7-scenes | 10 | 0.035 | 0.017 | 0.024 | 0.007 | 0.627   | 0.701    |
> | VGGT  | Global | NRGBD   | 20 | 0.012 | 0.007 | 0.014 | 0.004 | 0.893   | 0.985    |
> | StreamVGGT  | Streaming | NRGBD   | 20 | 0.088 | 0.055 | 0.051 | 0.016 | 0.753   | 0.932    |
> | **Ours**   | Streaming | NRGBD   | 20 | 0.073 | 0.034 | 0.026 | 0.008 | 0.758   | 0.917    |
>
> [1] Zhuo D, Zheng W, Guo J, et al. Streaming 4D Visual Geometry Transformer[J]. arXiv preprint arXiv:2507.11539, 2025.
>
> **3 - More ablation studies**
>
> Thanks for your suggestion.
> Table 5 in the paper compares the 3D reconstruction results with and without the 3D hierarchical position embedding ("w/o 3DHPE" means we remove this hierarchical position embedding in our pointer-image interaction during the training and evaluation).
> To further demonstrate the effectiveness of our 3D hierarchical position embedding, we conducted more comprehensive ablation studies for this module.
> Specifically, we implemented the 3D position embedding using a single frequency base (i.e., removing the hierarchical design), retrained the model, and evaluated its 3D reconstruction performance ("w/o hierarchical freq" in the table below).
> As the results show, **applying no embedding in the pointer-image interaction, using a single-frequency embedding, and adopting our 3D hierarchical embedding lead to progressively improved reconstruction performance**.
>
> | Method | Dataset | Acc ↓ | Comp ↓ | NC ↑ | Dataset | Acc ↓ | Comp ↓ | NC ↑ |
> |:------:|:-----:|:-----:|:-----:|:-----:|:-----:|:-------:|:--------:|:--------:|
> | Ours (w/o 3DHPE) | 7-scenes | 0.180 | 0.180 | 0.683 | NRGBD | 0.145 | 0.123   | 0.770    |
> | Ours (w/o hierarchical freq) | 7-scenes | 0.130 | 0.143 | 0.722 | NRGBD | 0.085 | 0.080   | 0.819    |
> | **Ours (w 3DHPE)** | 7-scenes | **0.124** | **0.139** | **0.725** | NRGBD | **0.079** | **0.073**   | **0.824**    |
>
> To further demonstrate the advantages of our explicit pointers, we provide additional ablations.
> Specifically, after encoding the current frame into new features, we directly stored them into the memory (removing any information related to explicit coordinates), retrained and evaluated the model ("No fusion" in the table below).
> Additionally, we applied KNN directly on the implicit features to fuse them with the features in the memory after encoding the current frame, retrained and evaluated the model ("Implicit fusion" in the table below).
> **The results below can further demonstrate the effectiveness of our explicit pointer memory when conducting streaming reconstruction**.
>
> | Method | Dataset | Acc ↓ | Comp ↓ | NC ↑ | Dataset | Acc ↓ | Comp ↓ | NC ↑ |
> |:------:|:-----:|:-----:|:-----:|:-----:|:-----:|:-------:|:--------:|:--------:|
> | No fusion | 7-scenes | 0.188 | 0.158 | 0.647 | NRGBD | 0.183 | 0.128   | 0.722    |
> | Implicit fusion | 7-scenes | 0.197 | 0.146 | 0.642 | NRGBD | 0.189 | 0.139   | 0.730   |
> | **Ours (Spatial-aware explicit fusion)** | 7-scenes | **0.124** | **0.139** | **0.725** | NRGBD | **0.079** | **0.073**   | **0.824**    |
>
>
> We hope the above response can help address your concerns. We are happy to answer any additional questions you may have.

---

> > ### Comment · Reviewer_8Vja · 2025-08-04
> >
> > Thanks for the additional experiments. Most of my concerns are addressed.

---

### Note · Authors · 2025-08-13

Dear PCs, SACs, ACs and all of our reviewers,

We first want to express our gratitude for your great efforts to our paper.
We have found all the reviews and discussions very helpful to improve our work, and we have answered and addressed most of the concerns.
We appreciate all of you for your comments highlighting the strengths of our work:

- An insightful analysis of various 3D reconstruction methods and well-structured writing from problem formulation to technical solutions.
- An innovative, conceptually sound, and experimentally validated explicit pointer memory to conduct the feed-forward streaming 3D reconstruction.
- A well-designed and theoretically-sound 3D hierarchical position embedding to promote the pointer-image interaction.

We have also addressed most of the concerns from the reviewers:
- More details to better explain our motivation and method design.
- More ablations across diverse settings to validate the effectiveness and robustness of our method.
- More detailed discussions and experimental comparisons with other existing methods (CUT3R, VGGT, StreamVGGT, MASt3R-SLAM, MegaSAM).

We also want to use this opportunity to briefly reply to reviewer JUWw's final request for our insights on the coordinate system challenge of this field (which was posted shortly after the discussion deadline and we apologize that we did not have enough time to provide a response).
This is a common problem in this field as mentioned by the reviewer, and we think removing the manual selection of a specific reference frame and allowing the model to adaptively select a reference frame is one potential solution.
This can allow more flexibility of the model outputs and may eliminate the bias prior introduced by the anchor frame.

Finally, we would like to thank all the reviewers recognizing our contribution to this emerging field (feed-forward streaming 3D reconstruction).
We want to emphasize that our work contributes to the community by **proposing an explicit pointer memory mechanism to improve the trade-off between long-term information preserving and memory compactness during streaming 3D reconstruction**.
We will release code, data, and checkpoints to facilitate further research.

Thanks again to all of you for your time and efforts.

Best regards,

Authors of Submission 6317

---

### Decision · Program_Chairs · 2025-09-17

**Decision:**

Accept (poster)

**Comment:**

Point3R proposes an explicit spatial pointer memory with 3D hierarchical positional embedding for streaming 3D reconstruction. Reviewers praised the clarity, motivation, and experimental results. However, concerns remain: comparisons against SLAM methods (e.g., MASt3R-SLAM) are insufficient; evidence suggests the proposed memory can degrade on long trajectories; and accuracy does not surpass systems that combine geometric priors with efficient back-end optimization. The camera-ready should add a focused discussion of these trade-offs and include targeted experiments, such as long-trajectory evaluations with loop closure and direct pose and mapping comparisons to MASt3R-SLAM.